# Interpreting Multi-Layer Transformers for In-Context Linear Regression with Varying Covariance

## Abstract

We study how multi-layer softmax-attention transformers perform in-context linear regression, focusing on the challenging setting where the covariate distribution varies across sequences. We show that multi-layer transformers substantially outperform single-layer models, demonstrating that depth is critical for sufficient expressivity. Through a novel probing methodology using Gaussian test data, we reveal that the transformer approximates linear regression by implementing a variant of the gradient descent algorithm. The parameters of this algorithm are dependent on model depth and data distribution, but insensitive to number of attention heads or sequence length, corresponding to a consistent diagonal structure in the learned weight matrices. Building on this insight, we show that by incorporating a chain-of-thought-style intermediate step, the transformer can solve in-context instrumental variable (IV) regression, achieving performance comparable to a two-stage least squares estimator. Our findings provide new evidence that depth enables transformers to learn sophisticated in-context algorithms, bridging the gap between empirical performance and interpretable algorithmic behavior.

## 1 Introduction

Transformer-based Large Language Models (LLMs) (Touvron et al., 2023a;b) have revolutionized the field of artificial intelligence. A key aspect of their success is in-context learning (ICL) (Bhattamishra et al., 2023; Park et al., 2024; Tang et al., 2023), a remarkable ability to solve novel tasks by conditioning on a few examples provided in the input, without any parameter updates. This emergent capability has spurred a wave of research (Brown et al., 2020; Dong et al., 2022) aimed at understanding the underlying mechanisms that allow LLMs to learn on the fly.

There has been a surge of interest in understanding ICL by studying how transformers solve linear regression tasks using synthetic data (Garg et al., 2022). In a common experimental setup, models are trained on sequences of input-output pairs $\{(x_i, y_i)\}_{i=1}^n$, where the input $x_i$ is sampled from a constant Gaussian distribution $\mathcal{N}(0, \Sigma)$ (Ahn et al., 2023; Zhang et al., 2024; Akyürek et al., 2022). Under this controlled setting, researchers have shown that transformers implicitly learn to approximate the pre-conditioned gradient descent (GD) estimator for solving linear regression problems (Ahn et al., 2023; He et al., 2025). However, this method proves fragile, as transformers trained on a constant input distribution fail to generalize to out-of-distribution (OOD) settings where the input data covariance varies across sequences (Von Oswald et al., 2023). This failure occurs because the learned parameters of these models encode the specific covariance structure of their training data as the pre-conditioning matrix, which is not robust to distributional shifts. Consequently, when the covariance changes, the model's performance may suffer.

In this paper, we study a more general and challenging linear regression ICL regime, where the **input covariance matrix varies for each sequence**. This setup substantially more difficult than those considered in prior work (Ahn et al., 2023; Von Oswald et al., 2023) where the test-time input distribution may not have been seen during training. Within this regime, we aim to address the following two fundamental questions:

*(1) Can a transformer be trained to solve linear regression in context with varying covariances?*
*(2) If yes, what linear regression algorithm does the transformer learn to implement?*

To address the first question, we train looped multi-layer softmax-attention transformers (Giannou et al., 2023; Fan et al., 2024) within our varying-covariance regime. Our findings indicate that the model's success is not guaranteed, but rather is critically dependent on its architecture. Specifically, we observe a **monotonic improvement** in ICL performance with **increasing model depth**. This trend, however, saturates beyond a certain depth (empirically, three or more layers), which underscores the necessity of sufficient model depth for effective task acquisition.

To address the second question, we analyze the learned weight matrices of the transformer, revealing that the model learns to implement a variant of the GD algorithm in this regime. This learned algorithm is composed of two distinct operations: a linear transformation of the covariates $x_i$, followed by a GD-like update on a newly formulated regression problem that utilizes the transformed covariates from the previous step. We discover that the algorithm's parameters, such as its step sizes, are intricately linked to both the input data distribution and the model's depth. Notably, in contrast to standard multi-step GD, the learned algorithm is non-convergent; the loss escalates rapidly beyond an optimal number of iterations. This finding suggests that the transformer employs a coarse, non-convergent approximation strategy, which is intrinsically tied to its architecture and the data distribution, to solve the linear regression task in this challenging regime.

Building on our findings for multi-layer transformer, we extend our analysis to more complex in-context tasks as an application, using in-context instrumental variable (IV) regression as a case study. The classical solution for IV regression is a two-stage least-squares (2SLS) estimator. In the first stage, to address the endogeneity of $x_i$, we regress it on an instrumental variable $z_i$ to obtain fitted values $\hat{x}_i$. In the second stage, we regress the outcome $y_i$ on these fitted values $\hat{x}_i$. In the in-context learning setting where an unseen IV regression problem is presented to the transformers in the test time, this two-stage process presents a unique challenge, as the distribution of $\hat{x}_i$ varies across sequences, which matches our regime in the linear regression task. We designed a multi-layer transformer with a Chain-of-Thought (CoT) (Wei et al., 2022) style that explicitly models this two-stage process. In our architecture, a single-layer transformer predicts $\hat{x}_i$ in a first stage and a multi-layer transformer then performs the $y_i$ on $\hat{x}_i$, with the two blocks trained jointly. With this training strategy, the model effectively decomposes the IV regression problem into its two constituent parts: the first-stage transformer learns a GD estimator, while the second stage learns the same iterative algorithm identified in our linear regression regime. Our empirical results show that the model's performance approaches that of the classical two-stage least squares estimator.

Our main contributions are the following:

- Through extensive experiments, we show that multi-layer softmax-attention transformers are sufficiently expressive to solve in-context linear regression, even when the covariance of the input distribution varies arbitrarily across sequences.

- Analysis of the learned weight patterns reveals that the transformer implicitly implements an iterative variant GD algorithm. This algorithm is non-convergent, and its parameters depend on model depth and data distribution.

- We extend these insights to in-context IV regression, demonstrating that a multi-layer transformer, prompted with a single CoT step, achieves performance comparable to a two-stage least squares estimator.

## 2 RELATED WORK

**In-Context Learning** Large language models (LLMs) (Floridi and Chiriatti, 2020; Achiam et al., 2023) owe much of their empirical success to in-context learning (ICL), their ability to adapt to new tasks at inference time using only a prompt of input–output examples without any parameter updates (Liu et al., 2021; Min et al., 2021; Nie et al., 2022; Bai et al., 2023a). Despite this progress, a unified theoretical understanding of ICL remains an active research topic.

Existing theoretical work has taken several approaches. One prominent branch interprets ICL as an implicit form of Bayesian inference implemented by the transformer's attention mechanism (Zhang et al., 2023; Jeon et al., 2024; Hu et al., 2024). Another analyzes how transformers can simulate learning algorithms, such as gradient descent or kernel methods, to solve ICL tasks (Bai et al., 2023b; Zhang et al., 2025; Nichani et al., 2024; Sheen et al., 2024; Giannou et al., 2024). Other studies investigate the function classes that transformers can represent or learn in context (Garg et al., 2022) or explore emergent behaviors and the transient nature of ICL capabilities (Singh et al.,

2023; Dai et al., 2022; Olsson et al., 2022). Our work contributes to this area by focusing on how multi-layer softmax attention theoretically and empirically enables in-context linear regression.

**In-Context Linear and IV Regression.** To better understand how transformers acquire ICL abilities, researchers have extensively studied simple, interpretable tasks like linear regression. The pioneering work by (Garg et al., 2022) empirically showed that transformers can achieve near Bayes-optimal performance on these tasks. Subsequent theoretical analyses (Ahn et al., 2023; Zhang et al., 2024; Von Oswald et al., 2023; Akyürek et al., 2022; Mahankali et al., 2023) revealed that linear transformers can implement optimization algorithms, such as gradient descent and its variants. Further studies have focused on more complex attention mechanisms with non-linear activations (He et al., 2025; Chen et al., 2024; Cheng et al., 2023; Cui et al., 2024; Fu et al., 2024).

Unlike many of the existing works that assume a fixed covariate data distributions across sequences, we focus on the more challenging regime where the data distribution varies among sequences. We also extend our findings to more complex tasks like IV regression, which has been investigated in other recent studies (Liang et al., 2024; Wang et al., 2024).

## 3 PRELIMINARY

In this study, we largely follow prior experimental setups (Garg et al., 2022; He et al., 2025), with the key difference that we allow the covariate distribution varies across sequences.

### 3.1 DATA DISTRIBUTION AND EMBEDDING

The linear regression task involves $L$ examples $\{(x_\ell, y_\ell)\}_{\ell \in [L]}$, where $x_\ell \in \mathbb{R}^d$ and $y_\ell \in \mathbb{R}$. The covariates $x_\ell$ in each sequence is drawn i.i.d. from $\mathsf{P}_x = \mathcal{N}(0, \Sigma)$. Here, distinct from previous studies, we consider a more challenging scenario where the $\Sigma$ varies across sequences during training. Specifically, we set $\Sigma = U \mathrm{diag}(\lambda_1, \ldots, \lambda_d) U^\top$, where $U$ is a rotation matrix satisfying $UU^\top = I_d$, and the eigenvalues $\lambda_i$ are sampled i.i.d. from $\mathrm{Unif}(\lambda_{\min}, \lambda_{\max})$. The label $y_\ell$ for each $x_\ell$ is given by $y_\ell = \beta^\top x_\ell + \epsilon_\ell$, where the true regression vector $\beta \in \mathbb{R}^d$ is drawn from $\mathsf{P}_\beta = \mathcal{N}(0, I_d)$ and the noise term $\epsilon_\ell$ is sampled from $\mathcal{N}(0, \sigma^2)$. A test input $x_q$ is independently drawn from a distribution that may differ from the training distribution $\mathcal{N}(0, \Sigma)$. To facilitate in-context learning, we construct an embedding matrix $Z_{\text{ebd}}^{(0)} \in \mathbb{R}^{(d+1) \times (L+1)}$ by concatenating the $L$ training pairs $\{(x_\ell, y_\ell)\}_{\ell \in [L]}$ with the test input $x_q$ and a placeholder zero for its response:

$$Z_{\text{ebd}}^{(0)} = [Z \; z_q] = \begin{bmatrix} x_1 & x_2 & \ldots & x_\ell & x_q \\ y_1 & y_2 & \ldots & y_\ell & 0 \end{bmatrix}. \tag{1}$$

### 3.2 TRANSFORMER ARCHITECTURE

**Multi-Head Softmax Attention.** In this study, we consider the sequence-to-sequence multi-head softmax attention with $H$ parallel heads, whose parameters $\theta = \{O^{(h)}, V^{(h)}, K^{(h)}, Q^{(h)}\}_{h \in [H]} \subseteq \mathbb{R}^{(d+1) \times (d+1)}$. The attention operation is then given by

$$\mathtt{Attn}_\theta(Z_{\text{ebd}}) = \sum_{h=1}^{H} O^{(h)} V^{(h)} Z_{\text{ebd}} \cdot \mathtt{smax} \circ \mathtt{msk}(Z_{\text{ebd}}^\top K^{(h)\top} Q^{(h)} Z_{\text{ebd}}), \tag{2}$$

where $\mathtt{smax}(\cdot)$ denotes the column-wise softmax operation and $\mathtt{msk}(\cdot)$ applies an element-wise causal mask that assigns $-\infty$ to the final-column entries of the attention scores, *i.e.,* $\mathtt{msk}(\cdot)_{ij} = I_d \cdot \mathbf{1}(j < L+1) - \infty \cdot \mathbf{1}(j = L+1)$.

We mainly use **looped multi-layer transformers with residual links** in our study. For several layers $T$, the transformer's output $Z_{\text{ebd}}^{(t)}$ at layer $t$ is formulated as follows:

$$Z_{\text{ebd}}^{(t)} = \mathtt{TF}_\theta(Z_{\text{ebd}}^{(t-1)}) = Z_{\text{ebd}}^{(t-1)} + \mathtt{Attn}_\theta(Z_{\text{ebd}}^{(t-1)}), \tag{3}$$

where $Z_{\text{ebd}}^{(t-1)}$ denotes the input embedding ($t-1$ layer's output) at layer $t$. From the transformer's final output matrix $Z_{\text{ebd}}^{(T)}$, we extract the last $(d+1, L+1)$-th entry as the prediction $\hat{y}_q \in \mathbb{R}$. The goal of $\hat{y}_q$ is to estimate $\mathbb{E}[\hat{y}_q | x_q] = \beta^\top x_q$.

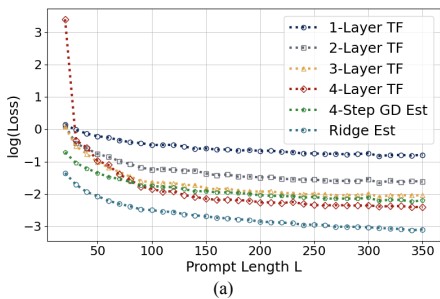 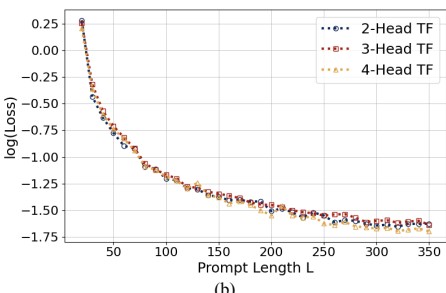

Figure 1: ICL prediction errors among: (a) transformer models and canonical estimators; (b) transformers with different heads.

**KQ and OV Circuits.** The computation in Equation (2) involves two components: computing the similarity between queries and keys, and using the outputs of the softmax function to generate the layer output. These two kinds of computation depend on matrix products $K^{(t,h)^\top} Q^{(t,h)}$ and $O^{(t,h)} V^{(t,h)}$ respectively, where $(t, h)$ represents the $t$-th layer and $h$-th head. To simplify the notation, we denote $K^{(t,h)^\top} Q^{(t,h)}$ as $\Phi^{(t,h)} \in \mathbb{R}^{(d+1)\times(d+1)}$, $O^{(t,h)} V^{(t,h)}$ as $\Psi^{(t,h)} \in \mathbb{R}^{(d+1)\times(d+1)}$. In addition, $\Phi^{(t,h)}$ and $\Psi^{(t,h)}$ are the same among different layer $t$ because of the looped transformer.

**Training Objective.** We train the model by minimizing the mean squared error, formulated as:

$$\mathcal{L}(\theta) = \mathbb{E}[(y_q - \hat{y}_q)^2] = \mathbb{E}[(y_q - \text{TF}_\theta(Z_{\text{ebd}}^{(0)}))^2]. \tag{4}$$

Optimization proceeds via mini-batch gradient-based algorithm (Adam (Kingma and Ba, 2014) in practice), with $\nabla \mathcal{L}(\theta)$ computed for every projection matrix in the parameter set $\theta$.

# 4 MECHANISTIC INTERPRETATION OF MULTI-LAYER TRANSFORMER

Recent most related works (Von Oswald et al., 2023; He et al., 2025) largely focus on single-layer models trained under a constant input distribution. Such assumptions limit understanding of how transformers generalize to covariates with different distributions. Here, we systematically examine multi-layer transformers trained on covariates with diverse covariance structures across sequences. We analyze their patterns and outputs, and show the algorithm learned by looped transformer.

## 4.1 EMPIRICAL RESULTS

**Experiment Setups.** For most experiments in the paper, we set the batch size $n_{\text{batch}} = 256$, the dimensions $d = 5$ of $x_\ell$, the sequence length $L = 120$, the covariance of the noise term $\sigma^2 = 0.1$, the minimum eigenvalue $\lambda_{\min} = 0.8$, and the maximum eigenvalue $\lambda_{\max} = 1.2$. We train the 2-layer, 3-layer, and 4-layer looped transformers with different numbers of heads for 100,000, 150,000, and 200,000 iterations, respectively. All weight matrices are initialized with small random values and are optimized by the Adam optimizer with a learning rate $10^{-3}$.

In the following, we summarize four main empirical findings from the experiments.

**Observation 1.** Transformer performance improves with increasing layers. Moreover, the marginal gains become negligible once the model comprises three or more layers, i.e., $T \geq 3$.

Figure 1 reports validation ICL loss across transformer depths. The single-layer model performs poorly under heterogeneous covariance due to the lack of expressivity. Adding layers steadily improves performance, but the benefit saturates: beyond three layers, additional depth brings almost no gain, with 3- and 4-layer curves nearly indistinguishable (by less than 0.01). In addition, the 4-layer transformer outperforms a 4-step standard GD estimator but remains inferior to the ridge estimator.

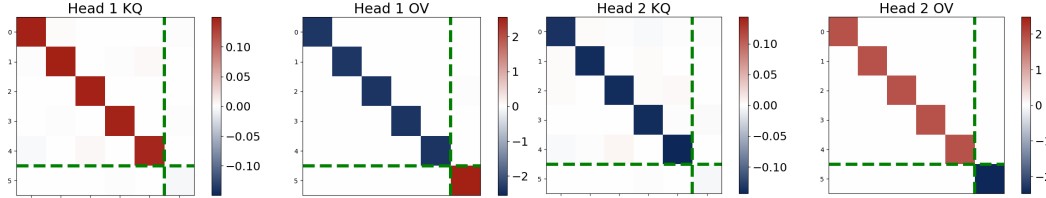

Figure 2: The learned patterns of 3-layer looped Transformer model. Notably, distinct patterns emerge in the KQ and OV circuits.

> **Observation 2.** For all multi-layer transformer models, Key-Query (KQ) and Output-Value (OV) circuits exhibit a consistent structure across layers and heads, which take the form of:
>
> $$\Phi^{(h)} = \begin{bmatrix} \omega^{(h)} I_d & 0_d \\ 0_d^\top & 0 \end{bmatrix}, \qquad \Psi^{(h)} = \begin{bmatrix} u_1^{(h)} I_d & 0_d \\ 0_d^\top & u_2^{(h)} \end{bmatrix}. \tag{5}$$
>
> where $I_d$ denotes a $d$-by-$d$ Identity matrix. Because we employ a looped transformer, the KQ and OV circuits are shared across layers; therefore, we omit the layer superscript $t$.

Figure 2 illustrates the patterns of KQ and OV circuit structures for the three-layer transformer with two attention heads. Across each layer, the diagonal entries of the learned KQ circuit are approximately constant, denoted by some $\omega^{(h)} \in \mathbb{R}$, with the exception of the final entry, which is zero. Similarly, in the OV circuit, the first $d$ diagonal entries are roughly equal to $u_1^{(h)} \in \mathbb{R}$ and the last entry is instead equal to $u_2^{(h)} \in \mathbb{R}$. These visualizations demonstrate that trained softmax attention transformers develop consistent and interpretable circuit patterns across layers.

> **Observation 3.** If $H \geq 2$, the performances of the transformer with the same layer are similar. The learned multi-head patterns exhibit the clear statistical regularities: $\langle u_1, \omega \rangle$ is negative, while $\langle u_2, \omega \rangle$ is positive.

Figure 1 (b) shows that all multi-head models achieve similar ICL errors, which suggest that multi-head transformers with $H \geq 2$ are equivalent to the two-head baseline.

> **Observation 4.** The trained transformer can generalize to the sequence length, however, when the transformer block is applied more than once during inference, the ICL prediction loss not only fails to decrease but in fact increases dramatically.

We train a 3-layer looped transformer and observe that its ICL prediction error increases rapidly as the number of inference steps grows from one to four. This suggests that multi-layer transformers implement a divergent iterative procedure, with each model optimized only for its own depth, which is different from the standard multi-step GD. We investigate the underlying mechanism in the next subsection.

More visualizations, including the empirical results for Observation 4, and experiments settings can be found in Appendix B.1.

### 4.2 INSIGHTS OF PATTERNS IN TRANSFORMER

As shown in Section 4.1, trained transformers display clear structural patterns in their KQ and OV circuits. By reparameterizing the model via the diagonal terms $(\omega^{(h)}, u_1^{(h)}, u_2^{(h)})$, we can interpret the algorithm learned by the transformer. This interpretation relies on the distribution transfer of the input sequences, rather than focusing on individual tokens.

We start with a general interpretation of the output by each attention layer through a Gaussian perspective. Note that this interpretation applies to a general transformer, independent of whether the weights are shared across layers. Accordingly, we include the superscript $t$ to denote layer-specific parameters.

**Claim 1.** Consider a $T$ layers, $H$ softmax attention heads transformer and let the input embedding $Z_{\text{ebd}}^{(0)} = (z_1^0, \ldots, z_\ell^0, z_q^0) \sim \mathcal{N}(\mu, \Sigma)$. If we assume the input sequence $L \to \infty$, the output $z_\ell^{(t+1)}$, at layer $t + 1$ can be represent as:

$$z_\ell^{(t+1)} = A^{(t)} z_\ell^{(t)} + b^{(t)}, \quad \forall (\ell, t) \in [L] \cup \{q\} \times [T-1],$$

$$\text{where} \quad A^{(t)} = I_d + \sum_{h=1}^{H} \Psi^{(t+1,h)} \Sigma^{(t)} \Phi^{(t+1,h)}, \quad b^{(t)} = \sum_{h=1}^{H} \Psi^{(t+1,h)} \mu^{(t)}, \tag{6}$$

$$\mu^{(t+1)} = A^{(t)} \mu^{(t)} + b^{(t)}, \quad \Sigma^{(t+1)} = A^{(t)} \Sigma^{(t)} A^{(t)^\top}, \quad \mu^{(0)} = \mu, \quad \Sigma^{(0)} = \Sigma.$$

This result offers an operator perspective of a multi-head attention layer. In particular, we can view the $t$-th multi-head attention layer as a mapping from the Gaussian distribution $\mathcal{N}(\mu^{(t-1)}, \Sigma^{(t-1)})$ to $\mathcal{N}(\mu^{(t)}, \Sigma^{(t)})$, and this mapping is parameterized by $\{\Psi^{(t,h)}, \Phi^{(t,h)}\}_{h \in [H]}$. The theoretical construction and experimental validation are provided in Appendix A.1. Throughout training, we set $L = 120$, which is sufficient to induce consistent structural patterns in the model. At each layer, the transformer maps the Gaussian input embedding $Z_{\text{ebd}}^{(0)}$ into a new distribution characterized by its mean $\mu$, covariance $\Sigma$ through the KQ circuit $\Phi$ and OV circuits $\Psi$. Our regime is a special case of the conditions in Equation (6), where the $\mu = 0$ and the $\Sigma$ is various across sequences.

Given the zero-mean inputs $z_\ell$, the transformer's output form, and the learned structured patterns in Equation (5), we establish the following iteration rule. Because our transformer is looped transformer, the parameter sets in KQ and OV circuits $\{u_1^{(t)}, u_2^{(t)}, \omega^{(t)}\}_{t=1}^{T}$ is the same regardless of the layer index $t$, we drop the superscript $t$ of $(u_1, u_2, \omega)$ for simplification from here.

**Claim 2.** *Given the learned patterns in the model, the layer-wise iteration rule $(x_\ell^{(t)}, y_\ell^{(t)}) \leftarrow (x_\ell^{(t+1)}, y_\ell^{(t+1)})$ for all $\ell \in [L] \cup \{q\}$ can be expressed as:*

$$x_\ell^{(t+1)} = x_\ell^{(t)} + \frac{\langle u_1, \omega \rangle}{L} \sum_{\ell'=1}^{L} x_{\ell'}^{(t)} x_{\ell'}^{(t)^\top} \cdot x_\ell^{(t)},$$

$$y_\ell^{(t+1)} = y_\ell^{(t)} + \frac{\langle u_2, \omega \rangle}{L} \sum_{\ell'=1}^{L} y_{\ell'}^{(t)} x_{\ell'}^{(t)^\top} \cdot x_\ell^{(t)}. \tag{7}$$

This update rule can be concluded in the following iterative Algorithm 4.2. In this algorithm, the step sizes $\eta_x$ and $\eta_y$ correspond to the absolute values of the inner products $\langle u_1, \omega \rangle$ and $\langle u_2, \omega \rangle$. The construction of this update rule can be found in Appendix A.2. Hereafter, we refer to this algorithm as *"interpreted algorithm"*.

---

**Algorithm 1** Interpreted Algorithm Implemented by Multi-Layer Transformer

---

**Require:** Step-sizes $\eta_x, \eta_y$, iterations time $T$
1: **for** each $t \in \{0, 1, \ldots, T-1\}$ and $\ell \in [L] \cup \{q\}$ **do**
2: $\quad x_\ell^{(t+1)} \leftarrow x_\ell^{(t)} - \eta_x/L \cdot \sum_{\ell'=1}^{L} x_{\ell'}^{(t)} x_{\ell'}^{(t)^\top} \cdot x_\ell^{(t)}$
3: $\quad y_\ell^{(t+1)} \leftarrow y_\ell^{(t)} + \eta_y/L \cdot \sum_{\ell'=1}^{L} y_{\ell'}^{(t)} x_{\ell'}^{(t)^\top} \cdot x_\ell^{(t)}$
4: **end for**
5: **return** $y_q^{(T)}$

---

We compare a three-layer, two-head softmax transformer with its interpreted algorithm, a three-step GD estimator, and the ridge regression estimator. Figure 3 (a) shows that the transformer and its interpreted algorithm produce nearly identical loss curves, outperforming three-step GD but falling short of ridge regression. Figure 3 (b) evaluates alignment between the transformer and the interpreted algorithm via two measures (Von Oswald et al., 2023): (1) cosine similarity between their input sensitivity vectors $\frac{\partial \hat{y}\text{TF}(x_q)}{\partial x_q}$ and $\frac{\partial \hat{y}\text{Alg}(x_q)}{\partial x_q}$ (Model Sim in the figure), and (2) the L2 norm of their difference, $\left\| \frac{\partial \hat{y}_{\text{TF}}(x_q)}{\partial x_q} - \frac{\partial \hat{y}_{\text{Alg}}(x_q)}{\partial x_q} \right\|_2$ (Grad Diff in the figure). These two metrics assess how similarly the transformer and the interpreted algorithm translate changes in $x_q$ into changes in the

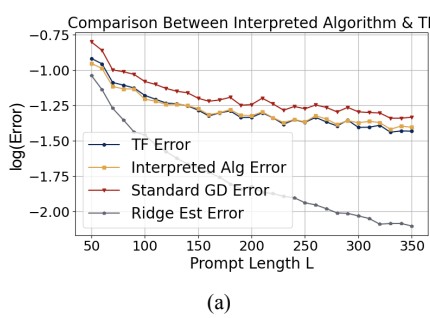
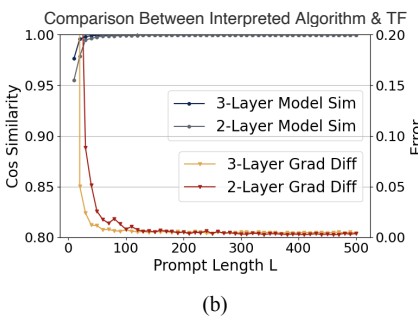

(a)                                                          (b)

Figure 3: (a) The performance comparison among 3-layer looped transformer, its interpreted algorithm, and 3-step GD; (b) the alignment between transformer and interpreted algorithm.

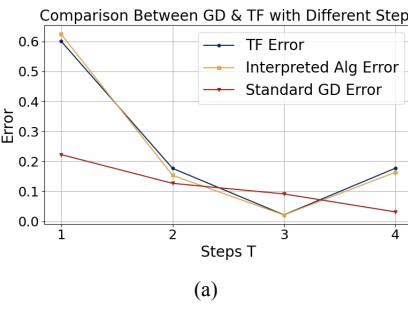
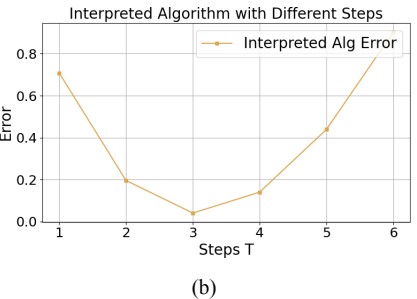

(a)                                                          (b)

Figure 4: (a) performance comparison on each step (layer for transformer) among a 3-layer looped transformer, its interpreted algorithm, and standard multi-step GD estimator; (b) performance of the 3-layer looped transformer's interpreted algorithm with varying numbers of steps.

prediction $\hat{y}_q$. Both metrics confirm a strong agreement: two- and three-layer models exhibit high cosine similarity and small L2 deviation.

These results demonstrate that the transformer effectively implements the analytically derived algorithm, with behavior nearly indistinguishable from our interpretation.

Below, we provide some additional discussions and properties of this interpreted algorithm.

**Entry-wise Homogeneous Transformation.** Our interpreted algorithm applies two homogeneous transformations to each input token pair $\{(x_\ell, y_\ell)\}_{\ell \in [L] \cup \{q\}}$. The first transformation is a linear update for the covariates $x_\ell^{(t)}$ with a negative step-size $-\eta_x$. The second transformation can be seen as a one-step GD on a newly constructed linear regression problem, with a learning rate $\eta_y$ and a residual link. This process updates the target variable $y_\ell^{(t+1)}$ for the $(t+1)$-th iteration, treating as a new linear regression task at $t$-th iteration solved via one-step GD with a residual link.

**Divergence with Increasing Steps.** We observed a divergent property in the multi-layer looped transformer (**Observation 4**), and we now demonstrate the same behavior in its interpreted algorithm. We conducted experiments on a 3-layer looped transformer to evaluate its convergence. As shown in Figure 4, the interpreted algorithm exhibits a divergent phenomenon that mirrors the transformer's behavior. This stands in sharp contrast to the standard multi-step GD estimator, which are guaranteed to converge with increasing step size.

**Comparison With Prior Work.** Prior work (Von Oswald et al., 2023; Ahn et al., 2023) has analyzed the behavior of multi-layer linear transformers. Von Oswald et al. (2023) studied inputs from $\mathcal{N}(0, I_d)$ and concluded that the model learns a GD algorithm, while Ahn et al. (2023) extended this to $\mathcal{N}(0, \Sigma)$ with constant covariance during training. These works interpreted the learned algorithm up to the model's depth. We extend this analysis by considering a more challenging setting with varying covariance across sequences and by analyzing the learned algorithm beyond the transformer's depth.

Further comparisons and discussions, including looped vs. original transformers, softmax vs. linear attention, single-layer vs. multi-layer architectures, and varying-covariance vs. constant-covariance regimes, are presented in Appendix C.

## 5 APPLICATION ON IV REGRESSION

Our findings on the capabilities of multi-layer transformers extend to more complex in-context learning tasks. We demonstrate this by successfully applying our approach to IV regression. We first show that IV regression presents a similar challenge to our varying covariate distribution setting, and then we demonstrate that the multi-layer transformer learns the interpreted algorithm for this scenario.

### 5.1 DATA GENERATION AND TRAINING SETTINGS

Our data generation process mainly follows that of (Liang et al., 2024). Suppose we wish to estimate the relationship between a scalar response variable $y_\ell \in \mathbb{R}$ and a $p$-dimensional predictor variable $x_\ell \in \mathbb{R}^p$. Unlike the ordinary-least-squares (OLS) setting, here the covariate $x_i$ is endogenous, *i.e.,* it is correlated with the regression noise $\epsilon_\ell$. Concretely, we observe $y_\ell = \beta^\top x_\ell + \epsilon_\ell$ and $x_\ell$ is endogenous. This correlation leads to a bias in the result when using OLS to estimate it. To address this endogeneity, we introduce an instrumental variable $z_\ell$ that is correlated with the endogenous variable $x_\ell$ but uncorrelated with the error term $\epsilon_\ell$. Under these assumptions, the instrument isolates the variation in $x_\ell$ to the omitted endogeneity, allowing for consistent inference on $\beta$.

With these insights, we generate $z_\ell \overset{\text{i.i.d.}}{\sim} \mathcal{N}(0, I_q)$, $u_\ell \overset{\text{i.i.d.}}{\sim} \mathcal{N}(0, \Sigma_u)$, $\omega_\ell \overset{\text{i.i.d.}}{\sim} \mathcal{N}(0, \Sigma_\omega)$, and then set

$$x_\ell = \theta^\top z_\ell + \Phi^\top u_\ell + \omega_\ell, \quad y_\ell = \beta^\top x_\ell + \phi^\top u_\ell + \epsilon_\ell, \tag{8}$$

so that $u_\ell$ serves as the common source of endogeneity between $x_\ell$ and $y_\ell$.

Given the observation values $\{(z_\ell, x_\ell, y_\ell)\}_{\ell \in [L]}$, the classic approach to estimate the IV regression can be divided into two stages. In the first stage, it regresses $x_\ell$ on $z_\ell$ to obtain predicted values $\hat{x}_\ell$. In the second stage, the method regresses $y_\ell$ on $\hat{x}_\ell$. Given that $\hat{x}_\ell$ in the second stage depends on the sequence-specific estimator $\hat{\theta}$, its distribution varies across sequences. This aligns with our regime in Section 3, highlighting the practical relevance of our covariate distribution setup.

**Sequence Generation.** In order to guide the transformer to implement the two-stage estimator, we decompose the procedure into two successive transformer blocks that are trained jointly. We define the stage 1 input as

$$Z_{\text{ebd}}^{\text{st1}} = \begin{bmatrix} z_1 & z_2 & \cdots & z_\ell & z_q \\ x_1 & x_2 & \cdots & x_\ell & 0 \end{bmatrix}, \tag{9}$$

and denote the first and second blocks by $\text{TF}_\theta^{\text{st1}}$ and $\text{TF}_\theta^{\text{st2}}$, respectively. The input sequence for the second stage $Z_{\text{ebd}}^{\text{st2}}$ is formed by concatenating of the entire output sequence of $\text{TF}_\theta^{\text{st1}}$ (with the gradient stop) with $\{y_\ell\}_{\ell \in [L] \cup \{q\}}$. Formally,

$$\{\hat{x}_\ell^{(\text{st1})}\}_{\ell \in [L] \cup \{q\}} = sg\big[\text{TF}_\theta^{\text{st1}}(Z_{\text{ebd}}^{\text{st1}})\big], \quad Z_{\text{ebd}}^{\text{st2}} = \text{cat}(\{\hat{x}_\ell^{(\text{st1})}\}_{\ell \in [L] \cup \{q\}}, \{y_\ell\}_{\ell \in [L] \cup \{q\}}), \tag{10}$$

where $sg[\cdot]$ denotes the gradient stop operation, and $\text{cat}(\cdot)$ denotes the concatenation.

**Training Settings.** The architectures of $\text{TF}_\theta^{\text{st1}}$ is a single-layer two-head transformer and $\text{TF}_\theta^{\text{st2}}$ is a three-layer two-head transformer. Notably, we train two transformer blocks together. We define the training loss as a two-part loss, which is formulated as:

$$\mathcal{L}(\theta) = \underbrace{\mathbb{E}[(x_q - \text{TF}_\theta^{\text{st1}}(Z_{\text{ebd}}^{\text{st1}}))]}_{\text{Stage 1}} + \underbrace{\mathbb{E}[(y_q - \text{TF}_\theta^{\text{st2}}(Z_{\text{ebd}}^{\text{st2}}))]}_{\text{Stage 2}}. \tag{11}$$

More training details and pseudo-code for sequence generation can be found in Appendix B.2

### 5.2 EMPIRICAL RESULTS OF IV REGRESSION

Under our settings, a transformer trained with the two-stage strategy, similar in spirit to the chain-of-thought step, can effectively solve IV regression tasks. In the first stage, a single-layer transformer learns a GD estimator (He et al., 2025; Von Oswald et al., 2023) and outputs the fitted regressors $\{\hat{x}_\ell\}_{\ell \in [L] \cup \{q\}}$ based on the instruments $\{z_\ell\}_{\ell \in [L] \cup \{q\}}$ and endogenous variables $\{x_\ell\}_{\ell \in [L] \cup \{q\}}$. In the second stage, a multi-layer transformer learns the algorithm described in Section 4 and predicts the target $\{y_q\}$ using the fitted regressors $\{\hat{x}_\ell\}_{\ell \in [L] \cup \{q\}}$ together with $\{y_\ell\}_{\ell \in [L]}$.

This subsection focuses on the empirical observations of the transformer in the second stage. With the architecture and loss function described earlier, the results closely match those in Section 4.

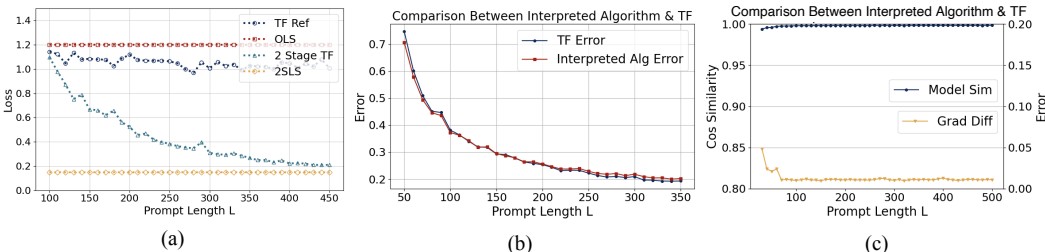

(a)          (b)          (c)

Figure 5: Results for IV regression tasks: (a) performance comparison among canonical estimators, transformer trained to regress $y_\ell$ on $x_\ell$ (TF Ref), transformer trained in a two stage manner (2 Stage TF); (b) performance comparison with interpreted algorithm in the second stage; (c) alignment between the transformer and the interpreted algorithm.

> **Observation 5.** The full performance of the two-block transformer is close to a two-stage least square estimator, better than the OLS estimator and same layer transformers trained in one stage.

Figure 5 (a) shows the transformer's performance converges to that of the two-stage least squares estimator as the sample size $L$ increases. To confirm that our performance improvement is due to our two-stage training procedure and not simply increased model depth, we compared our approach to a single-stage baseline (TF Ref) where a transformer of the same depth directly regresses $y_\ell$ on $x_\ell$. As shown in Figure 5 (a), the transformer trained with our two-stage procedure significantly outperforms the single-stage baseline. The second stage requires a multi-layer transformer because the intermediate variable $\hat{x}_\ell$ depends on the sequence-specific estimator $\hat{\theta}$, which causes the covariance of $\hat{x}_\ell$ to vary across sequences. This dynamic covariance structure matches the problem setting analyzed in Section 3, highlighting the realistic application of our approach.

> **Observation 6.** The patterns of the KQ and OV circuits in the $\text{TF}_\theta^{\text{st2}}$ match the structure described in **Observation 2** in Section 4.

The learned KQ and OV circuits in the second stage, see in Figure 13, indicate that the transformer block in the second stage implements the same procedure as Algorithm 4.2. We further assess the alignment between the trained transformer and the interpreted algorithm, as well as their performance, in Figure 5 (b) and (c). These empirical results provide additional validation of our claim.

Consequently, the multi-layer transformer successfully learns the interpreted algorithm and accurately predicts the target $\{y_q\}$ using the fitted regressors $\{\hat{x}_\ell\}_{\ell \in [L] \cup \{q\}}$ together with $\{y_\ell\}_{\ell \in [L]}$. This demonstrates the capacity of multi-layer transformers to solve complex tasks.

## 6 CONCLUSION

We investigate the ability of transformers to perform in-context linear regression under varying input covariances, a scenario that moves beyond the fixed-distribution assumptions of prior work. Our experiments demonstrate that sufficiently deep transformers can generalize across heterogeneous covariances and reliably solve regression tasks in context. Through an analysis of the internal computations, we find that transformers implement an iterative algorithm that provides a depth-dependent approximation to the input data distribution.

We also extend these insights to IV regression, where a transformer prompted with a single chain-of-thought step replicates the performance of a two-stage least square. This shows that transformers can learn not only simple regressors but also multi-stage statistical pipelines entirely in context.

These findings underscore the algorithmic generalization capacity of multi-layer transformers under distributional shift and open avenues for studying in-context learning in more challenging domains.

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

## A DERIVATIONS OF METHODOLOGY IN SECTION 4

### A.1 MECHANISTIC INTERPRETATION VIA GAUSSIAN POPULATION

Consider a transformer with testing input consisting of i.i.d. Gaussian in-context samples, i.e., $Z^{(0)} = [z_1, \ldots, z_L] \in \mathbb{R}^{d \times L}$ where $z_\ell^{(0)} \overset{\text{i.i.d.}}{\sim} \mathcal{N}(\mu, \Sigma)$, along with a given query $z_q$. We remark that our setup falls under this framework, as the parameter $\beta$ remains fixed across all in-context samples during the testing stage. Here, we focus on the regime where $L \to \infty$. Note that $z_\ell^{(t)}$ are identically distributed due to the i.i.d sampling scheme of $Z$ and the nearly position-invariant transformation within the attention-only transformer architecture under large $L$.

In the following, we conduct a induction argument under the limiting regime. Recall that we assume $z_\ell^{(0)} \overset{\text{i.i.d.}}{\sim} \mathcal{N}(\mu, \Sigma)$ to be the data generation distribution of input. Next, suppose $z_\ell^{(t)} \overset{\text{i.i.d.}}{\sim} \mathcal{N}(\mu_t, \Sigma^{(t)}) := p_t$ for all $\ell \in [L] \cup \{q\}$. For each layer iteration $t$, we have

$$z_\ell^{(t+1)} = z_\ell^{(t)} + \sum_{h=1}^{H} \Psi^{(t+1,h)} Z_{1:L}^{(t)} \cdot \texttt{smax}(Z_{1:L}^{(t)\top} \Phi^{(t+1,h)} z_\ell^{(t)})$$

$$\approx z_\ell^{(t)} + \sum_{h=1}^{H} \int_{\mathbb{R}^d} \frac{\Psi^{(t+1,h)} z' \cdot \exp(z'^\top \Phi^{(t+1,h)} z_\ell^{(t)}) \cdot p_t(z')}{\int_{\mathbb{R}^d} \exp(z''^\top \Phi^{(t+1,h)} z_\ell^{(t)}) \cdot p_t(z'') \, \mathrm{d}z''} \, \mathrm{d}z'$$

$$= z_\ell^{(t)} + \sum_{h=1}^{H} \int_{\mathbb{R}^d} \frac{\Psi^{(t+1,h)} z' \cdot \exp(z'^\top \Phi^{(t+1,h)} z_\ell^{(t)} - \|z' - \mu^{(t)}\|_{\Sigma^{(t)-1}}^2/2)}{\int_{\mathbb{R}^d} \exp(z''^\top \Phi^{(t+1,h)} z_\ell^{(t)} - \|z'' - \mu^{(t)}\|_{\Sigma^{(t)-1}}^2/2) \, \mathrm{d}z''} \, \mathrm{d}z'$$

$$= z_\ell^{(t)} + \sum_{h=1}^{H} \mathbb{E}_{z' \sim \tilde{p}_t}[\Psi^{(t+1,h)} z'], \tag{12}$$

where the distribution $\tilde{p}_t$ is given by

$$\tilde{p}_t(z') \propto \exp(z'^\top \Phi^{(t+1,h)} z_\ell^{(t)} - \|z' - \mu^{(t)}\|_{\Sigma^{(t)-1}}^2/2) = \mathcal{N}(\mu^{(t)} + \Sigma^{(t)} \Phi^{(t+1,h)} z_\ell^{(t)}, \Sigma^{(t)}), \tag{13}$$

and $Z_{1:L}^{(t)}$ means the set of $\{z_i\}_{i=1}^{L}$. Here, the approximation results from the law of large number (LLN) for sufficiently large $L$:

$$\Psi^{(t+1,h)} Z_{1:L}^{(t)} \cdot \texttt{smax}(Z_{1:L}^{(t)\top} \Phi^{(t+1,h)} z_\ell) = \sum_{\iota=1}^{L} \Psi^{(t+1,h)} z_\iota \cdot \frac{\exp(z_\iota^\top \Phi^{(t+1,h)} z_\ell^{(t)})}{\sum_{\iota'=1}^{L} \exp(z_{\iota'}^\top \Phi^{(t+1,h)} z_\ell^{(t)})}$$

$$= \frac{\frac{1}{L} \sum_{\iota=1}^{L} \Psi^{(t+1,h)} z_\iota \cdot \exp(z_\iota^\top \Phi^{(t+1,h)} z_\ell^{(t)})}{\frac{1}{L} \sum_{\iota'=1}^{L} \exp(z_{\iota'}^\top \Phi^{(t+1,h)} z_\ell^{(t)})} \approx \frac{\mathbb{E}_{z_\iota \sim p_t}[\Psi^{(t+1,h)} z_\iota \cdot \exp(z_\iota^\top \Phi^{(t,h)} z_\ell^{(t)})]}{\mathbb{E}_{z_{\iota'} \sim p_t}[\exp(z_{\iota'}^\top \Phi^{(t+1,h)} z_\ell^{(t)})]}.$$

Because the causal mask prohibits the query token $z_q^{(t)}$ from attending to itself, its iterative update likewise satisfies (12). Combining (12) and (13), after the $t$-th layer iteration, the output distribution is given by

$$z_\ell^{(t+1)} = z_\ell^{(t)} + \sum_{h=1}^{H} \Psi^{(t+1,h)} \big(\mu^{(t)} + \Sigma^{(t)} \Phi^{(t+1,h)} z_\ell^{(t)}\big)$$

$$= \Big(I_d + \sum_{h=1}^{H} \Psi^{(t+1,h)} \Sigma^{(t)} \Phi^{(t+1,h)}\Big) z_\ell^{(t)} + \sum_{h=1}^{H} \Psi^{(t+1,h)} \mu^{(t)} := A^{(t)} z_\ell^{(t)} + b^{(t)}. \tag{14}$$

Hence, from a population perspective, the transformer conducts a local linear transformation of $z_\ell^{(t)}$, characterized by a tranformation matrix $A^{(t)}$ and a bias term $b^{(t)}$. Moreover, given (14) and fact that $z_\ell^{(t)} \overset{\text{i.i.d.}}{\sim} \mathcal{N}(\mu^{(t)}, \Sigma^{(t)})$, it holds that

$$p_{t+1} = \mathcal{N}(A^{(t)} \mu^{(t)} + b^{(t)}, A^{(t)\top} \Sigma^{(t)} A^{(t)})$$

$$\text{s.t.} \quad \mu^{(t+1)} = A^{(t)} \mu^{(t)} + b^{(t)} \quad \text{and} \quad \Sigma^{(t+1)} = A^{(t)} \Sigma^{(t)} A^{(t)\top}, \tag{15}$$

which completes the induction argument. To summarize, by combining (14) and (15), thanks to the gaussian samples and large $L$, we can interpret the transformer via a recursive form as below:

$$z_\ell^{(t+1)} = A^{(t)} z_\ell^{(t)} + b^{(t)}, \quad \forall (\ell, t) \in [L] \cup \{q\} \times [T-1],$$

$$\text{where} \quad A^{(t)} = I_d + \sum_{h=1}^{H} \Psi^{(t+1,h)} \Sigma^{(t)} \Phi^{(t+1,h)}, \quad b^{(t)} = \sum_{h=1}^{H} \Psi^{(t+1,h)} \mu^{(t)},$$

$$\mu^{(t+1)} = A^{(t)} \mu^{(t)} + b^{(t)}, \;\; \Sigma^{(t+1)} = A^{(t)} \Sigma^{(t)} A^{(t)^\top}, \;\; \mu^{(0)} = \mu, \;\; \Sigma^{(0)} = \Sigma. \quad (16)$$

We evaluate this interpretation by comparing the transformer's output to the theoretical results of Equation (16). Specifically, we analyze a three-layer transformer with random initialization, using inputs $z_i$ sampled from a random Gaussian distribution $\mathcal{N}(\mu, \Sigma)$. As shown in Figure 6, we observe that the empirical output of the transformer closely aligns with the theoretical values as the sequence length $L$ increases. This strong agreement validates our recursive interpretation. Our regime in the main paper is a specific case, where $\mu = 0$ and the covariance $\Sigma$ varies across sequences.

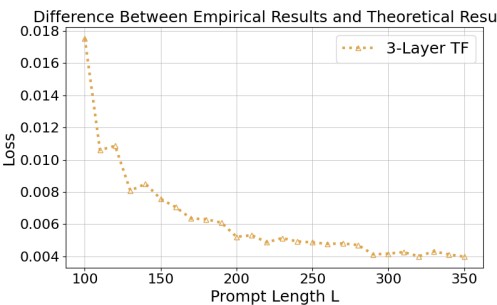

Figure 6: The MSE loss between the empirical transformer output and theoretical transformer output.

## A.2 SIMPLIFICATION UNDER EMERGED PATTERNS

Recall that the learned looped transformer satisfies the following patterns:

$$\Phi^{(h)} = \begin{bmatrix} \omega^{(h)} I_d & 0_d \\ 0_d^\top & 0 \end{bmatrix}, \quad \Psi^{(h)} = \begin{bmatrix} u_1^{(h)} I_d & 0_d \\ 0_d^\top & u_2^{(h)} \end{bmatrix}, \quad \forall (h) \in [H]. \quad (17)$$

Denote $\omega = (\omega^{(1)}, \ldots, \omega^{(H)}) \in \mathbb{R}^H$, $u_1 = (u_1^{(1)}, \ldots, u_1^{(H)}) \in \mathbb{R}^H$, and $u_2 = (u_2^{(1)}, \ldots, u_2^{(H)}) \in \mathbb{R}^H$. Moreover, under the embedding and data generation process in Section 3, we have

$$z_\ell = [x_\ell \; y_\ell]^\top \in \mathbb{R}^{d+1}, \quad \text{where} \;\; x_\ell \overset{\text{i.i.d.}}{\sim} \mathcal{N}(0_d, \Sigma), \;\; y_\ell = \langle x_\ell, \beta \rangle + \epsilon_\ell \; \text{with} \; \epsilon \overset{\text{i.i.d.}}{\sim} \mathcal{N}(0, \sigma^2).$$

Let $\Gamma$ denote the covariance matrix of $z_\ell$. Hence, by simple calculation, we have $\mu = \mathbb{E}[z_\ell] = 0$ and

$$\Gamma = \begin{bmatrix} \Gamma_{11} & \Gamma_{12} \\ \Gamma_{12}^\top & \Gamma_{22} \end{bmatrix} \in \mathbb{R}^{(d+1) \times (d+1)},$$

$$\text{where} \;\; \Gamma_{11} = \Sigma \in \mathbb{R}^{d \times d}, \quad \Gamma_{12} = \Sigma \beta \in \mathbb{R}^d, \quad \Gamma_{22} = \|\beta\|_\Sigma^2 + \sigma^2 \in \mathbb{R}.$$

Next, we simplify the recuisive form in (16) based on the specific parametrization above. Note that $\mu_0 = 0_{d+1}$. Suppose that $\mu^{(t)} = 0_{d+1}$, by a simple induction argument, it holds that

$$b^{(t)} = \sum_{h=1}^{H} \Phi^{(h)} \mu^{(t)} = 0_{d+1} \; \text{s.t.} \; \mu^{(t+1)} = A^{(t)} \mu^{(t)} + b^{(t)} = 0_{d+1},$$

which implies that $\mu^{(t)} = 0_{d+1}$ for all $t \in [T]$. Then, we can reduce (16) to

$$z_\ell^{(t+1)} = A^{(t)} z_\ell^{(t)}, \quad \forall (\ell, t) \in [L] \cup \{q\} \times [T-1],$$

$$\text{where} \quad A^{(t)} = I_d + \sum_{h=1}^{H} \Psi^{(h)} \Gamma^{(t)} \Phi^{(h)}, \quad \Gamma^{(t+1)} = A^{(t)} \Gamma^{(t)} A^{(t)^\top} \; \text{with} \; \Gamma^{(0)} = \Gamma. \quad (18)$$

Next, we derive the recursive form of covariance. First, based on (17) and (18), note that

$$A^{(t)} = I_{d+1} + \sum_{h=1}^{H} \Psi^{(h)} \Gamma^{(t)} \Phi^{(h)} := \begin{bmatrix} A_{11}^{(t)} & 0_d \\ A_{21}^{(t)} & 1 \end{bmatrix}$$

$$\text{where} \quad A_{11}^{(t)} = I_d + \sum_{h=1}^{H} u_1^{(h)} \omega_1^{(h)} \cdot \Gamma_{11}^{(t)} \quad \text{and} \quad A_{21}^{(t)} = \sum_{h=1}^{H} u_2^{(h)} \omega_1^{(h)} \cdot \Gamma_{12}^{(t)}{}^{\top}.$$

Using this block decomposition in the layer-wise iteration rule (16), we obtain

$$z_\ell^{(t+1)} = \begin{bmatrix} x_\ell^{(t+1)} \\ y_\ell^{(t+1)} \end{bmatrix} = A^{(t)} \begin{bmatrix} x_\ell^{(t)} \\ y_\ell^{(t)} \end{bmatrix} = \begin{bmatrix} x_\ell^{(t)} \\ y_\ell^{(t)} \end{bmatrix} + \begin{bmatrix} \langle u_1, \omega \rangle \cdot \Gamma_{11}^{(t)} \cdot x_\ell^{(t)} \\ \langle u_2, \omega \rangle \cdot \Gamma_{12}^{(t)}{}^{\top} \cdot x_\ell^{(t)} \end{bmatrix}$$

$$= \begin{bmatrix} x_\ell^{(t)} \\ y_\ell^{(t)} \end{bmatrix} + \begin{bmatrix} \frac{\langle u_1, \omega \rangle}{L} \sum_{\ell'=1}^{L} x_{\ell'}^{(t)} x_{\ell'}^{(t)}{}^{\top} \cdot x_\ell^{(t)} \\ \frac{\langle u_2, \omega \rangle}{L} \sum_{\ell'=1}^{L} y_{\ell'}^{(t)} x_{\ell'}^{(t)}{}^{\top} \cdot x_\ell^{(t)} \end{bmatrix}, \tag{19}$$

when $L$ is large enough.

## B  MORE TRAINING DETAILS AND SETTINGS

### B.1  EXTENSIVE EXPERIMENTS ON LINEAR REGRESSION TASK

In this subsection, we conduct experiments with single head transformer, increased dimensionality, and a broader range of covariance eigenvalues to evaluate the robustness and consistency of our conclusions. Furthermore, we provide some visualizations which do not provided in the main paper due to the limited space. The corresponding results are presented below.

**More KQ and OV Visualizations.** We extend our analysis beyond the three-layer transformer presented in the main paper by providing KQ and OV visualizations for two- and four-layer models, see in Figure 7. These visualizations reveal that the diagonal patterns learned by the model are highly consistent across different depths, indicating a generalizable learning mechanism. This finding reinforces our main conclusion that transformers learn a specific algorithmic structure, regardless of the number of layers.

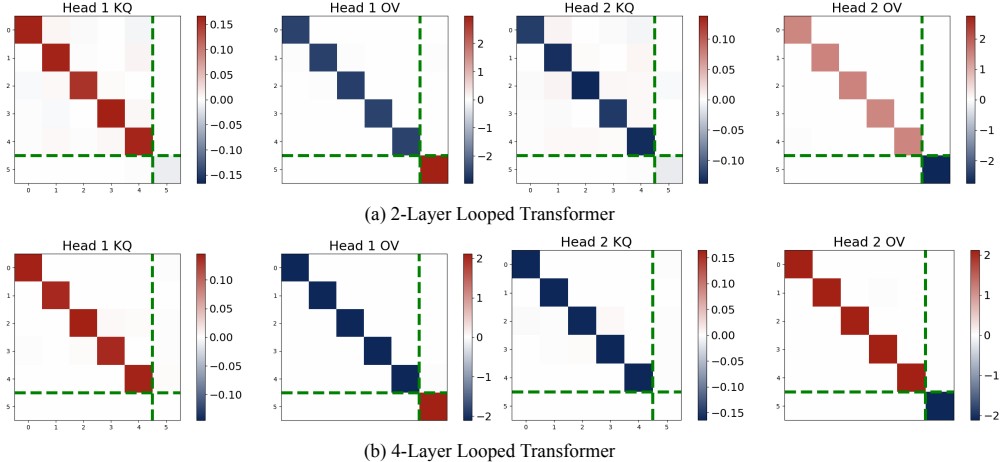

Figure 7: The learned patterns of 2- and 4-layer looped Transformer model.

**Performance Comparison with Absolute Scale.** We plot the absolute performance differences among the 1-, 2-, 3-, and 4-layer transformers in Figure 8. The figure clearly shows a substantial gap between the 1-layer transformer and the multi-layer models. Moreover, when the number of layers exceeds three, the additional performance improvement becomes negligible (less than 0.01).

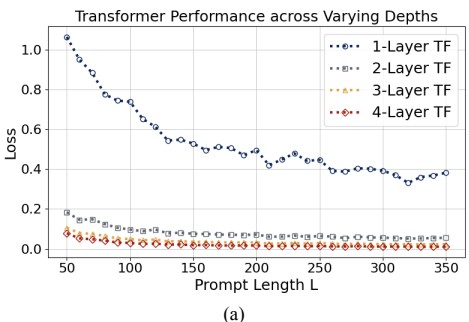

(a)

Figure 8: The learned patterns of 2- and 4-layer looped Transformer model.

**Performance of Single-head Transformers.** We also conduct experiments on transformers with a single attention head. As shown in Figure 9, we use a two-layer model as an example, other settings are the same with those in main paper. We observe that the learned KQ and OV circuit patterns are structurally similar to those in multi-head models but the learned inner products $\langle u_1, \omega \rangle$ and $\langle u_2, \omega \rangle$ are different. This inner products difference leads to inferior performance, as seen in Figure 9 (b). This phenomenon is consistent with findings from He et al. (2025), which noted that single-head attention corresponds to a less expressive, non-parametric predictor. Our results demonstrate that this limitation persists in multi-layer transformers, leading to their reduced performance.

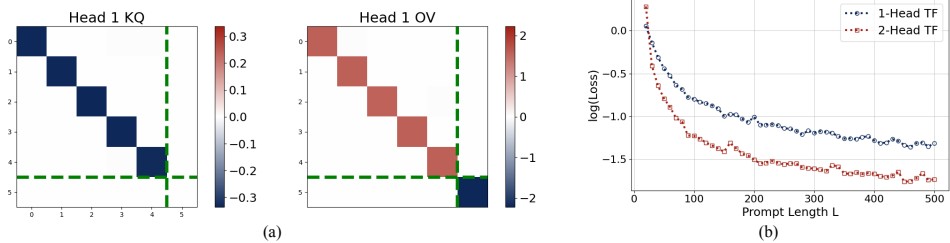

Figure 9: The results of the single head transformer: (a) The patterns of KQ and OV circuits; (b) ICL loss comparison between single-head and multi-head.

**Performance of Stacked Transformers.** We provide the performance of stacked transformers in Figure 10. When the transformer is forward propagated multiple times, the ICL error increases rapidly. This suggests the model learns a non-convergent algorithm, where the algorithm's behavior is dependent on the transformer's depth, as discussed in Section 4.

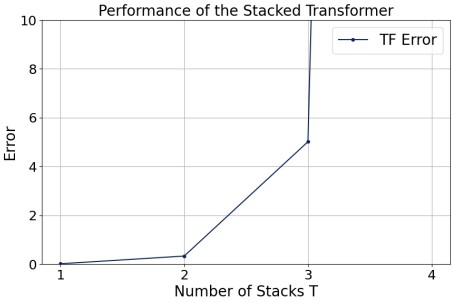

Figure 10: Performance with different stack depths.

**Larger Dimension.** Figure 11 shows the performance and learned patterns of a 3-layer, 2-head transformer with dimensionality $d = 12$. The KQ and OV circuit patterns remain consistent with those observed in the $d = 5$ case (Figure 11 (a)). Moreover, the transformer aligns closely with the

interpreted algorithm (Figure 11 (b), (c)), with alignment evaluated as in Section 4. These results demonstrate that our conclusions generalize robustly to higher-dimensional settings.

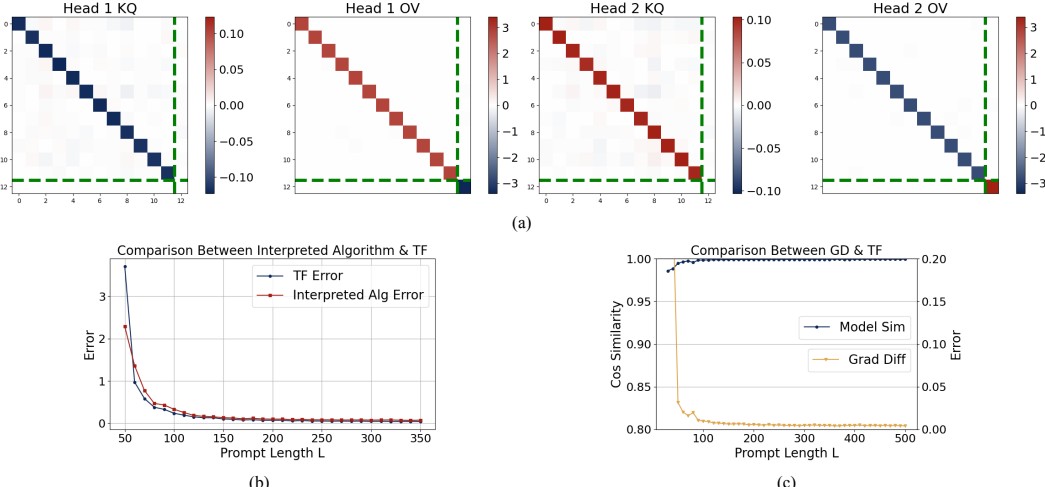

(a)

(b)

(c)

Figure 11: (a) The learned patterns of a 3-layer, 2-head Transformer model when the dimension $d = 12$. (b) Performance comparison between the transformer and the interpreted algorithm. (c) Alignment between the transformer and the interpreted algorithm.

**Broader Range of Covariance Eigenvalues.** We further examine the impact of varying the covariance eigenvalues $\lambda_i$ used in data generation by considering a wider range, specifically setting $\lambda_{\min} = 0.5$ and $\lambda_{\max} = 3$. A 3-layer, 2-head transformer is used for this experiment. Figure 12 (a) visualizes the learned KQ and OV circuits, (b) compares the performance of the trained transformer and the interpreted algorithm, and (c) quantifies their alignment. These results provide additional validation for the proposed interpreted algorithm and lead to conclusions consistent with those obtained under the eigenvalue range considered in the main paper, which shows the robustness of interpreted algorithm on a broader range of covariance eigenvalues.

## B.2 Training Details for the IV Regression Task

**Sequence Generation Details.** In this section, we describe the process of generating training sequences for the IV regression task. Recall that the input sequences in stage 1 and stage 2 are $\{(z_\ell, x_\ell)\}_{\ell \in [L] \cup \{q\}}$ and $\{(\hat{x}_\ell, y_\ell)\}_{\ell \in [L] \cup \{q\}}$, respectively. The variables $x_\ell$ and $y_\ell$ are computed according to Equation (8). During training, $x_q$ and $y_q$ are replaced with zero tokens. A key step in this two-stage training procedure is obtaining $\{\hat{x}_\ell\}_{\ell \in [L] \cup \{q\}}$. For this purpose, we employ a single-layer, two-head softmax attention transformer in the first stage. Since the task of this stage is to regress $\hat{x}_\ell$ on $z_\ell$, which corresponds to a linear regression task with covariates sampled from a standard Gaussian distribution. The transformer learns a GD predictor once the sequence length $L$ is sufficiently large (He et al., 2025) on this linear regression task. Specifically, for all $x_\ell \in [L] \cup \{q\}$, we have $x_\ell = \frac{1}{L} \sum_{\ell'}^{L} x_{\ell'} z_{\ell'}^{\top} \cdot z_\ell$. Thus, the fitted regressors $\hat{x}_\ell$ can be obtained without the influence of the noise terms $u_\ell$ and $w_\ell$, by applying the trained transformer from the first stage. In practice, unlike prior work that only considers the query position as the output, we take the entire sequence as the output. Meanwhile, each input token is prevented from attending to itself during inference through a causal mask. However, during stage 1 training, we still compute the loss only on the final token, as defined in Equation (11). The pseudo-code for sequence generation and training IV regression tasks is provided in Algorithm B.2.

**Hyper-parameters Details.** In this part, we describe the training configuration for the in-context IV regression task. For the experiment in Section 5, we set the instrument dimension of $z_\ell$ to $q = 6$ and the endogenous variable dimension of $x_\ell$ to $p = 4$. The $\Sigma_u$, $\Sigma_\omega$, and $\Sigma_\epsilon$ are all Identity matrix. The sequence length is $L = 200$, and the model is trained for 200,000 iterations. The first stage transformer is a single-layer transformer, and the second stage transformer is a three-layer looped transformer. All other hyper-parameters remain the same as those specified in Section 4.

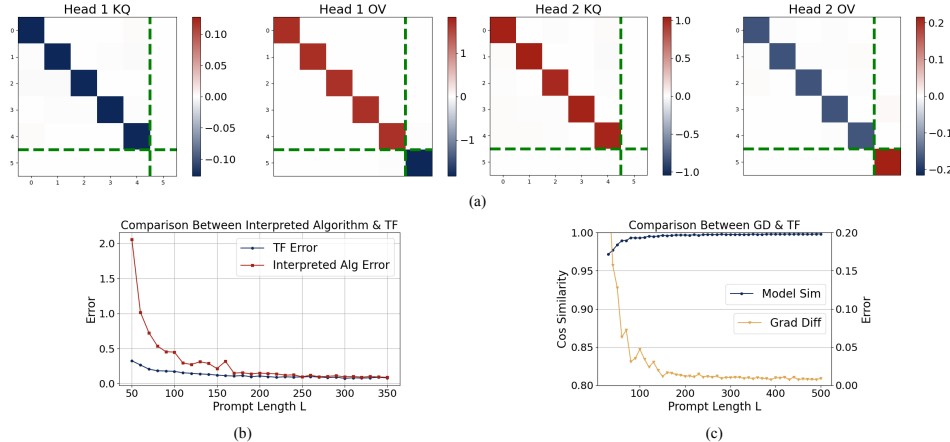

(a)

(b)                                              (c)

Figure 12: (a) The learned patterns of a 3-layer, 2-head Transformer model when the range of eigenvalues is broader. (b) Performance comparison between the transformer and the interpreted algorithm. (c) Alignment between the transformer and the interpreted algorithm.

---

**Algorithm 2** One Iteration for Training In-Context IV Regression Tasks

---
**Require:** Sequence Length $L$, Task parameters $\theta, \Phi, \beta, \phi$, Transformer Blocks $\text{TF}_\theta^{\text{st1}}, \text{TF}_\theta^{\text{st2}}$.
1: **for** $\ell = 1, \ldots, L$ **do**                                        ▷ Data generation
2:    **Sample:** $z_\ell \overset{\text{i.i.d.}}{\sim} \mathcal{N}(0, I_q), u_\ell \overset{\text{i.i.d.}}{\sim} \mathcal{N}(0, \Sigma_u), \omega_\ell \overset{\text{i.i.d.}}{\sim} \mathcal{N}(0, \Sigma_\omega), \epsilon_\ell \overset{\text{i.i.d.}}{\sim} \mathcal{N}(0, \Sigma_\epsilon)$
3:    **Compute:** $x_\ell = \theta^\top z_\ell + \Phi^\top u_\ell + \omega_\ell$
4:    **Compute:** $y_\ell = \beta^\top x_\ell + \phi^\top u_\ell + \epsilon_\ell$                        ▷ According to (8)
5: **end for**
6: **Sample:** $z_q \overset{\text{i.i.d.}}{\sim} \mathcal{N}(0, I_q), \omega_\ell \overset{\text{i.i.d.}}{\sim} \mathcal{N}(0, \Sigma_\omega), \epsilon_\ell \overset{\text{i.i.d.}}{\sim} \mathcal{N}(0, \Sigma_\epsilon)$
7: **Compute:** $x_q = \theta^\top z_q$
8: **Compute:** $y_q = \beta^\top x_q$                                         ▷ Generate query token
9: **Concatenate:** $Z_{\text{ebd}}^{\text{st1}} \leftarrow \{(\{z_\ell\}_{\ell \in [L]}, z_q), (\{x_\ell\}_{\ell \in [L]}, 0)\}$
10: **Inference:** $(\{\hat{x}_\ell\}_{\ell \in [L]}, \hat{x}_q) \leftarrow \text{TF}^{\text{st1}}$                        ▷ Gradient stop
11: **Concatenate:** $Z_{\text{ebd}}^{\text{st2}} \leftarrow \{(\{\hat{x}_\ell\}_{\ell \in [L]}, \hat{x}_q), (\{y_\ell\}_{\ell \in [L]}, 0)\}$
12: **Calculate:** loss $= \mathcal{L}_{mse}(x_q, \text{TF}_\theta^{\text{st1}}(Z_{\text{ebd}}^{\text{st1}})) + \mathcal{L}_{mse}(y_q, \text{TF}_\theta^{\text{st2}}(Z_{\text{ebd}}^{\text{st2}}))$
13: **Train:** loss.backward(), optimizer.step()
14: **return** $\text{TF}_\theta^{\text{st1}}, \text{TF}_\theta^{\text{st2}}$

---

**Learned Patterns of KQ and OV Circuit.** We visualized the learned KQ and OV patterns of the second-stage transformer in Figure 13. The patterns are similar to those learned in our linear regression experiments with varying covariate distributions, suggesting that the transformer learns the same interpreted algorithm to solve the second stage of the IV regression task. In Section 5, we evaluate the alignment between the transformer's behavior and our interpreted algorithm. Our experimental results show a high degree of alignment, suggesting the multi-layer transformer can be applied to more complex scenarios.

## C  DISCUSSIONS AND EXTENSIONS

### C.1  LOOPED TRANSFORMER VS. ORIGINAL TRANSFORMER

In this subsection, we study the connection between looped transformer and the transformer without weight tying (original transformer). The experimental settings for original transformer are identical to those described in Section 4 but without weight tying of each layer, and we use the 2-layer transformer as a representative example. We observe two distinct behaviors depending on the random initialization: **1)** both the KQ and OV patterns are diagonal and resemble Equation 5 (*Structure* in the following text); **2)** the KQ and OV patterns appear disordered (*Disordered* in the following text).

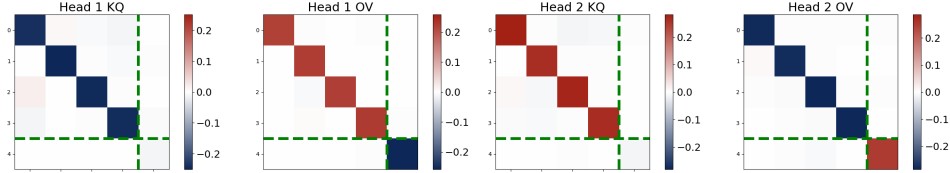

Figure 13: Visualizations of KQ and OV circuit patterns of the transformer in the second stage.

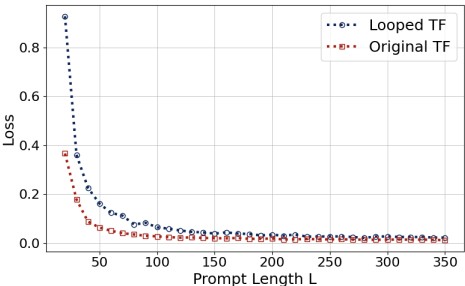

Figure 14: Performance comparison between looped and original transformer.

The corresponding patterns and ICL prediction performance are shown in Figure 15. When the KQ and OV patterns are diagonal, the learned behavior can be easily interpreted as Algorithm 4.2 with layer-dependent step sizes $\{\eta_x^t\}_{t=1}^T$ and $\{\eta_y^t\}_{t=1}^T$ (By adding the layer superscript $t$ to $\Phi$ and $\Psi$ in each layer). Figure 15 (d) provides empirical evidence to support this interpretation. However, when the KQ and OV patterns are disordered, we surprisingly find that the transformer still achieves strong ICL performance, see in Figure 15 (c). This suggests that the multi-layer transformer can successfully handle linear regression tasks even when the learned patterns deviate from a strictly diagonal structure.

We attribute this phenomenon to the increased representational flexibility of multi-layer transformers without weight tying. Such models can implement multiple distinct solution strategies for the linear regression task, with the choice of strategy heavily influenced by initialization. Certain random seeds lead to diagonal patterns and recovery of Algorithm 4.2, whereas others result in alternative algorithmic solutions. In contrast, the multi-layer looped transformer, due to its shared weights across layers, has a more constrained hypothesis space and consistently converges to Algorithm 4.2, regardless of initialization.

In addition, we compare the validation ICL loss of the looped and original transformers in Figure 14. Their performances converge as the sequence length $L$ increases, indicating that weight tying does not negatively impact transformer performance. The multi-layer looped transformer therefore retains sufficient expressivity to solve the linear regression task under our regime. Moreover, its simplified structure enables it to effectively learn the constant algorithm (the interpreted algorithm) required for this task.

### C.2 SOFTMAX ATTENTION VS. LINEAR ATTENTION

Previous works (Von Oswald et al., 2023; Ahn et al., 2023) have investigated the mechanisms of linear attention transformers on linear regression tasks with a constant input covariance $\Sigma$. This architecture, which removes the nonlinear activation, can be simplified and formulated as:

$$\text{TF}_\theta(Z_{\text{ebd}}) = Z_{\text{ebd}} + \frac{1}{L}\sum_{h=1}^H \Psi^{(h)} Z_{\text{ebd}} \cdot M \cdot Z_{\text{ebd}}^\top \Phi^{(h)} Z_{\text{ebd}}, \quad M = \begin{bmatrix} I_d & 0 \\ 0 & 0 \end{bmatrix}, \tag{20}$$

where $M$ is the causal mask, serving the same role as $\text{msk}(\cdot)$ in Equation 2, and $L$ represents the sequence length. Prior studies demonstrated that a single linear attention layer can implement one step of GD, while a multi-layer linear transformer can implement a GD++ algorithm when the input covariance is the identity matrix (Von Oswald et al., 2023).

Table 1: The inner-products of each layers.

| | Looped Softmax TF | | |
|---|---|---|---|
| | 2 Layers | 3 Layers | 4 Layers |
| $\langle u_1, \omega \rangle$ | -0.6675 | -0.6128 | -0.6590 |
| $\langle u_2, \omega \rangle$ | 0.8591 | 0.7193 | 0.6563 |
| | Looped Linear TF | | |
| | 2 Layers | 3 Layers | 4 Layers |
| $\langle u_1, \omega \rangle$ | -0.6616 | -0.6122 | -0.6466 |
| $\langle u_2, \omega \rangle$ | 0.8479 | 0.7214 | 0.6217 |

In this study, we extend this analysis to looped multi-layer linear transformers, investigating their behavior under a varying input covariance regime and establishing a connection between linear and softmax transformers. We adapt the linear attention architecture from (Ahn et al., 2023), with a key modification: we set all attention parameters to be trainable, unlike the original work which fixes the final entries of the KQ and OV circuits. This guarantees the same structure and training objective between linear and softmax attention.

The input data construction and training settings follow the methodology outlined in Sections 3 and 4, respectively. Using a 2-layer, 1-head linear transformer as a representative example, we observe that the model develops structural patterns similar to those observed with softmax transformers in the main paper, as shown in Figure 16 (a). Furthermore, the ICL loss of the linear transformer closely aligns with that of its softmax counterpart, as illustrated in Figure 16 (b).

With the learned patterns, we have the multi-layer linear transformer iteration rules for $\ell \in [L] \cup \{q\}$:

$$
z_\ell^{(t+1)} = z_\ell^{(t)} + \frac{1}{L} \sum_{h=1}^{H} \Psi^{(h)} Z_{1:L}^{(t)} \cdot (Z_{1:L}^{(t)\top} \Phi^{(h)} z_\ell^{(t)})
$$

$$
= \begin{bmatrix} x_\ell^{(t)} \\ y_\ell^{(t)} \end{bmatrix} + \frac{1}{L} \sum_{h=1}^{H} \begin{bmatrix} u_1^{(h)} I_d & 0_d \\ 0_d^\top & u_2^{(h)} \end{bmatrix} \cdot \begin{bmatrix} X_{1:L}^{(t)} \\ Y_{1:L}^{(t)} \end{bmatrix} \cdot \begin{bmatrix} X_{1:L}^{(t)} \\ Y_{1:L}^{(t)} \end{bmatrix}^\top \cdot \begin{bmatrix} \omega^{(h)} I_d & 0_d \\ 0_d^\top & 0 \end{bmatrix} \cdot \begin{bmatrix} x_\ell^{(t)} \\ y_\ell^{(t)} \end{bmatrix}
$$

$$
= \begin{bmatrix} x_\ell^{(t)} \\ y_\ell^{(t)} \end{bmatrix} + \frac{1}{L} \sum_{h=1}^{H} \begin{bmatrix} u_1^{(h)} X_{1:L}^{(t)} X_{1:L}^{(t)\top} & u_1^{(h)} X_{1:L}^{(t)} Y_{1:L}^{(t)\top} \\ u_2^{(h)} Y_{1:L}^{(t)} X_{1:L}^{(t)\top} & u_2^{(h)} Y_{1:L}^{(t)} Y_{1:L}^{(t)\top} \end{bmatrix} \cdot \begin{bmatrix} \omega^{(h)} x_\ell^{(t)} \\ 0 \end{bmatrix} \tag{21}
$$

$$
= \begin{bmatrix} x_\ell^{(t)} \\ y_\ell^{(t)} \end{bmatrix} + \frac{1}{L} \begin{bmatrix} \langle u_1, \omega \rangle X_{1:L}^{(t)} X_{1:L}^{(t)\top} x_\ell^{(t)} \\ \langle u_2, \omega \rangle Y_{1:L}^{(t)} X_{1:L}^{(t)\top} x_\ell^{(t)} \end{bmatrix}
$$

$$
= \begin{bmatrix} x_\ell^{(t)} \\ y_\ell^{(t)} \end{bmatrix} + \frac{1}{L} \begin{bmatrix} \langle u_1, \omega \rangle \sum_{\ell'=1}^{L} x_{\ell'}^{(t)} x_{\ell'}^{(t)\top} \cdot x_\ell^{(t)} \\ \langle u_2, \omega \rangle \sum_{\ell'=1}^{L} y_{\ell'}^{(t)} x_{\ell'}^{(t)\top} \cdot x_\ell^{(t)} \end{bmatrix},
$$

where $Z_{1:L}^{(t)}$, $X_{1:L}^{(t)}$, $Y_{1:L}^{(t)}$ means $\{z_\ell^{(t)}\}_{\ell=1}^{L}$, $\{x_\ell^{(t)}\}_{\ell=1}^{L}$, and $\{y_\ell^{(t)}\}_{\ell=1}^{L}$ respectively. We find that the update rule learned by the multi-layer linear-attention transformer is identical to Equation (7), confirming that the model implements Algorithm 4.2. As shown in Figure 17 (a) and (b), the linear transformer's prediction loss and its interpreted algorithm are nearly identical. In addition, the alignment of two models also high. Both of the evidence provides strong empirical support for our theoretical analysis. We observed similar behavior in non-looped linear transformers (Figure 18). However, the patterns in these non-looped models are inconsistent and depend on initialization, similar to their softmax counterparts. The learned structure patterns confirm that the model implements Algorithm 4.2.

In addition, as reported in Table 1, the learned inner products $\langle u_1, \omega \rangle$ and $\langle u_2, \omega \rangle$ at each layer in the looped linear model are nearly identical to those found in looped softmax transformers. The learned step sizes $\eta_x$ and $\eta_y$ also match, suggesting that in the varying covariance setting, trained linear transformers are functionally equivalent to softmax transformers for solving linear regression tasks.

## C.3 VARYING COVARIANCE VS. CONSTANT COVARIANCE

In this subsection, we analyze how transformers behave when trained on input distribution with constant covariance, a setting that extends the work of (Ahn et al., 2023; Von Oswald et al., 2023)

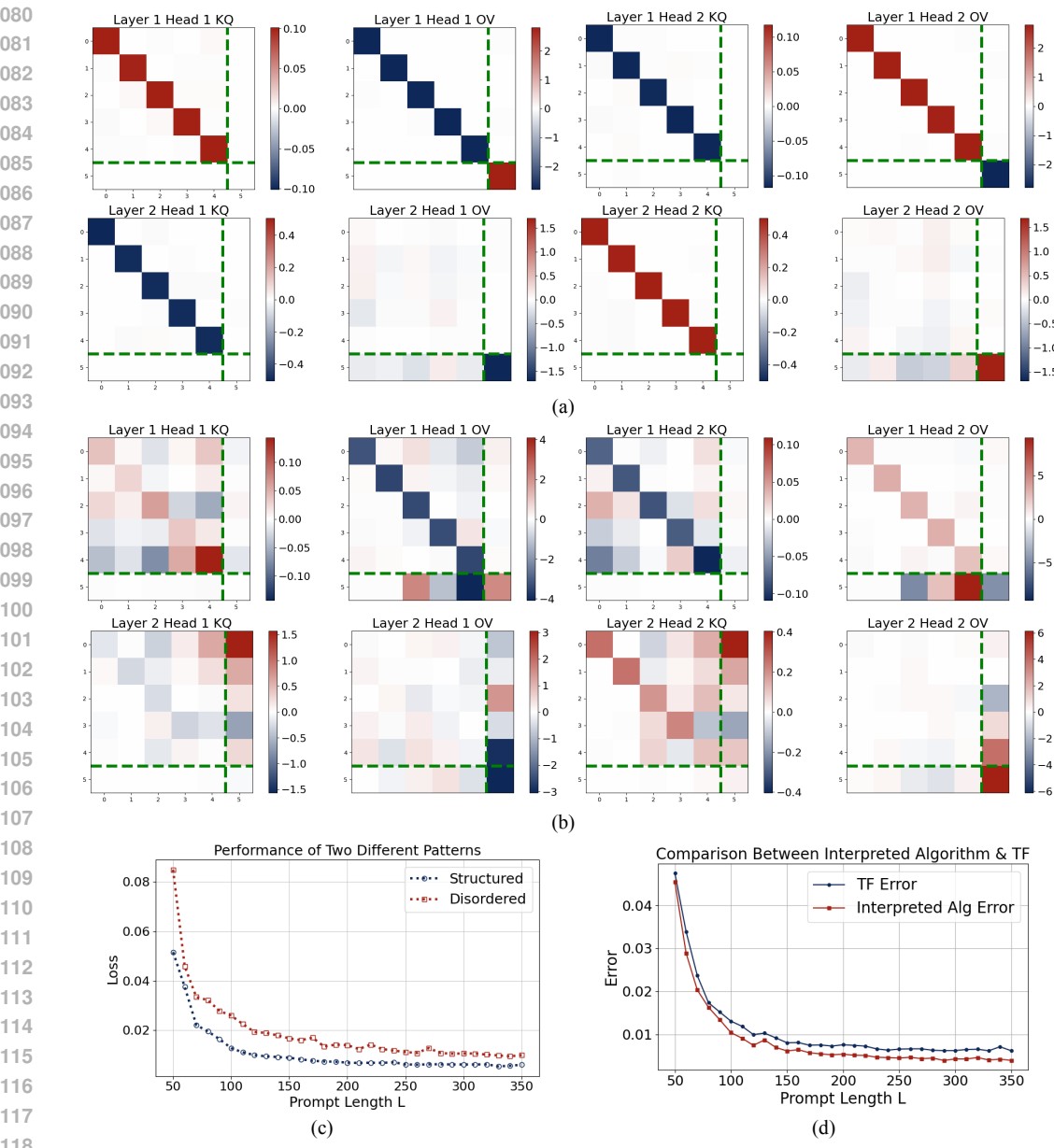

Figure 15: Comparison of two pattern types: (a) structured patterns in a 2-layer, 2-head transformer; (b) disordered patterns in the same architecture with different initialization; (c) ICL prediction loss for structured vs. disordered patterns; (d) performance of the structured-pattern transformer vs. its interpreted algorithm.

to softmax attention models. In addition, we study the generalization ability of models trained on varying covariance when tested on a constant covariance distribution.

**Transformer trained with constant covariance.** We first consider the case where covariates are sampled from $\mathcal{N}(0, \Sigma)$ with fixed covariance matrix $\Sigma$ for each sequence. Two settings are examined: (1) $\Sigma$ as the identity matrix, and (2) $\Sigma$ as a KMS matrix (Fikioris, 2018) with parameter $\rho = 0.5$, whose $(i, j)$-entry is $\Sigma_{ij} = \rho^{|i-j|}$. We train a 2-layer, 2-head looped transformer under the same experimental setup as in Section 4, and have the following observation.

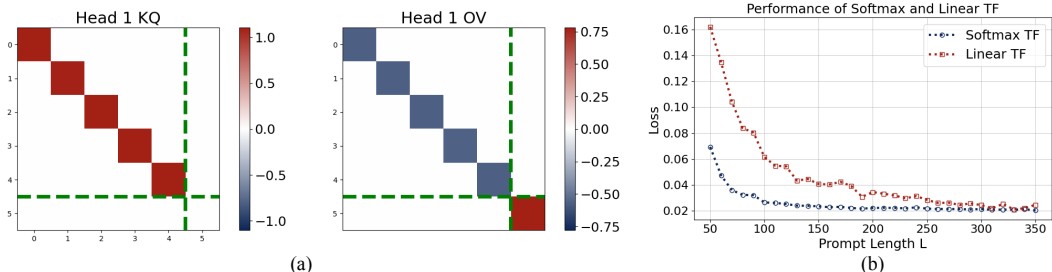

(a)                    (b)

Figure 16: Results for a 2-layer, 1-head linear transformer: (a) KQ and OV pattern visualizations; (b) performance comparison with a softmax transformer of the same depth.

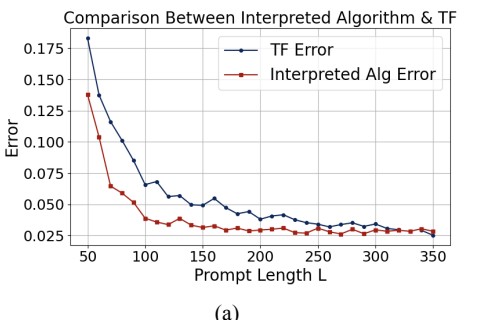
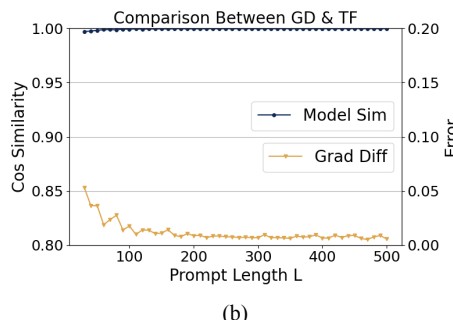

(a)                    (b)

Figure 17: Results for a 2-layer 1-head linear transformer: (a) Performance comparison between the transformer and the interpreted algorithm. (b) Alignment between the transformer and the interpreted algorithm.

**Observation.** Transformers encode covariance structure into KQ circuits when the covariance of $x_\ell$ is constant.

The learned patterns differ from those reported in Section 4 and show strong dependence on $\Sigma$. Figures 20 (a) and (b) display the KQ and OV patterns for two cases, respectively. When the covariates are anisotropic, the KQ patterns are no longer diagonal, although the OV patterns remain consistent with Equation (5). The learned structure can be expressed as:

$$\Phi^{(h)} = \begin{bmatrix} \omega^{(h)}\Omega I_d & 0_d \\ 0_d^\top & 0 \end{bmatrix}, \qquad \Psi^{(h)} = \begin{bmatrix} u_1^{(h)} I_d & 0_d \\ 0_d^\top & u_2^{(h)} \end{bmatrix}, \tag{22}$$

where $\Omega$ is a $d$-by-$d$ preconditioning matrix. Prior work (He et al., 2025) shows that in single-layer transformers with sufficiently long sequences, $\Omega$ approximates $\Sigma^{-1}$. Our experiments confirm that this result also holds in the multi-layer setting.

Figure 20 (c) visualizes $\Sigma^{-1}$, revealing a close correspondence between $\Sigma^{-1}$ and the learned KQ patterns. Figure 20 (d) plots the L2 distance between each KQ circuit multiply the covariance matrix $\Phi \cdot \Sigma$ and a scaled identity matrix. The distance decreases during training and converges to a small value, indicating strong alignment.

These findings suggest that the transformer implements an alternative update rule:

$$x_\ell^{(t+1)} = x_\ell^{(t)} + \frac{\langle u_1, \omega \rangle \cdot \Sigma^{-1}}{L} \sum_{\ell'=1}^L x_{\ell'}^{(t)} x_{\ell'}^{(t)\top} \cdot x_\ell^{(t)},$$

$$y_\ell^{(t+1)} = y_\ell^{(t)} + \frac{\langle u_2, \omega \rangle \cdot \Sigma^{-1}}{L} \sum_{\ell'=1}^L y_{\ell'}^{(t)} x_{\ell'}^{(t)\top} \cdot x_\ell^{(t)}. \tag{23}$$

Here, $u_1$ is defined as the mean of the diagonal elements of $\Phi \cdot \Sigma$, excluding the last entry.

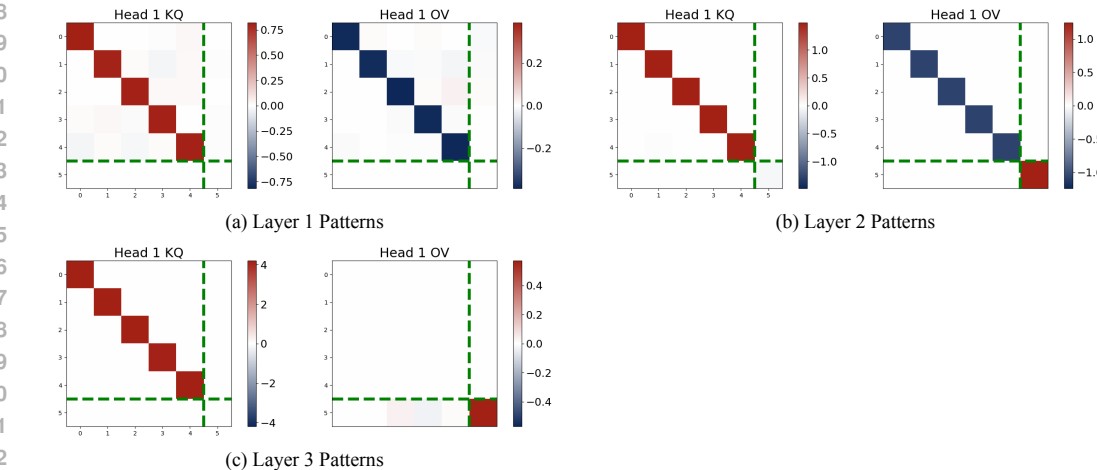

(a) Layer 1 Patterns

(b) Layer 2 Patterns

(c) Layer 3 Patterns

Figure 18: Visualization for a 3-layer 1-head linear transformer with structured patterns.

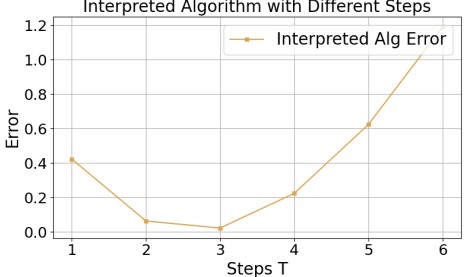

Figure 19: ICL error of the interpreted algorithm with varying number of steps.

Figure 20 (e) compares the performance of the trained transformer and the interpreted algorithm. The covariance matrix of the input data distribution is a KMS matrix. Their results nearly overlap, validating the correctness of our interpretation. We conclude that multi-layer transformers can effectively handle covariates with constant covariance by adapting the learned algorithm to incorporate the covariance structure, which is similar to the single-layer case.

**Generalization of the learned algorithm with constant covariance.** We also test the depth generalization of the algorithm in this regime, as shown in Figure 19. We find that this algorithm is also non-convergent and that the learned parameters are highly correlated with the model's depth. This observation is consistent with the findings in our main paper, suggesting that, regardless of the input distribution, a multi-layer transformer learns a non-convergent algorithm that approximates linear regression in a way that is highly dependent on its architecture.

**Transformer inference on constant covariance.** We next examine how a transformer trained on varying covariance generalizes to test data with constant covariance. Specifically, we compare two multi-layer transformers: one trained on varying covariance and the other on constant covariance. The test data are sampled from a Gaussian distribution with identity covariance matrix.

As shown in Figure 20 (e), both models perform well. The transformer trained on varying covariance achieves performance only slightly worse (by about 0.04) than the model trained directly on constant covariance. This result demonstrates the flexibility of transformers across different input distributions. Since constant covariance is a special case of varying covariance, the model trained on the broader distribution retains strong generalization.

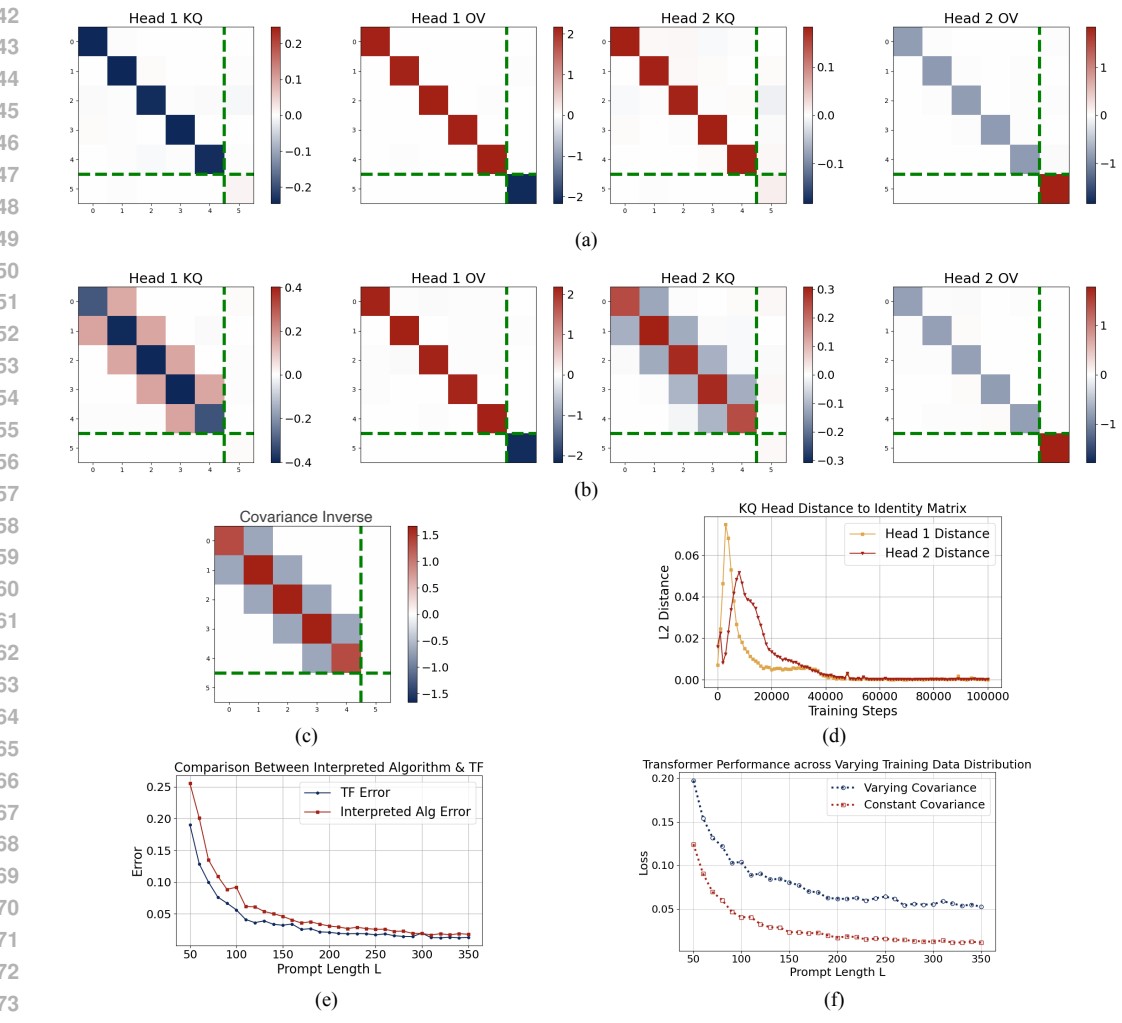

Figure 20: Results for transformers trained with different input distributions: (a) KQ and OV patterns for covariates from $\mathcal{N}(0, I_d)$; (b) KQ and OV patterns for covariates from $\mathcal{N}(0, \Sigma)$ with $\Sigma$ a KMS matrix ($\rho = 0.5$); (c) visualizations of $\Sigma^{-1}$; (d) L2 distance between the $\Phi \cdot \Sigma^{-1}$ and $u_1 \cdot I_d$; (e) performance comparison between transformer and interpreted algorithm; (f) performance comparison between transformers trained on different input distributions.

## C.4 SINGLE-LAYER TRANSFORMER VS. MULTI-LAYER TRANSFORMER

In this subsection, we analyze the differences between the KQ and OV circuits learned by a single-layer transformer versus a multi-layer transformer. As shown in Figure 21 (a), when a single-layer model is applied to linear regression with varying covariance structures, the learned patterns exhibit slight differences. Specifically, the first $d$ diagonal entries in OV circuits are close to zero. Given the model's single-layer architecture, the last column of the KQ circuit and the first $d$ columns of the OV circuit do not contribute to the final prediction $\hat{y}_q$. Similarly, the residual link has no effect on the prediction because the last entry of the input is set to zero. Consequently, the actual single-layer transformer output can be expressed as:

$$\hat{y}_q = \frac{\langle u_2, \omega \rangle}{L} \sum_{\ell'=1}^{L} y_{\ell'}^{(t)} x_{\ell'}^{(t)^\top} \cdot x_\ell^{(t)}. \tag{24}$$

This formulation reveals that the single-layer transformer learns an algorithm similar to Algorithm 4.2 but performs only a single iteration. This single-step limitation leads to its inferior per-

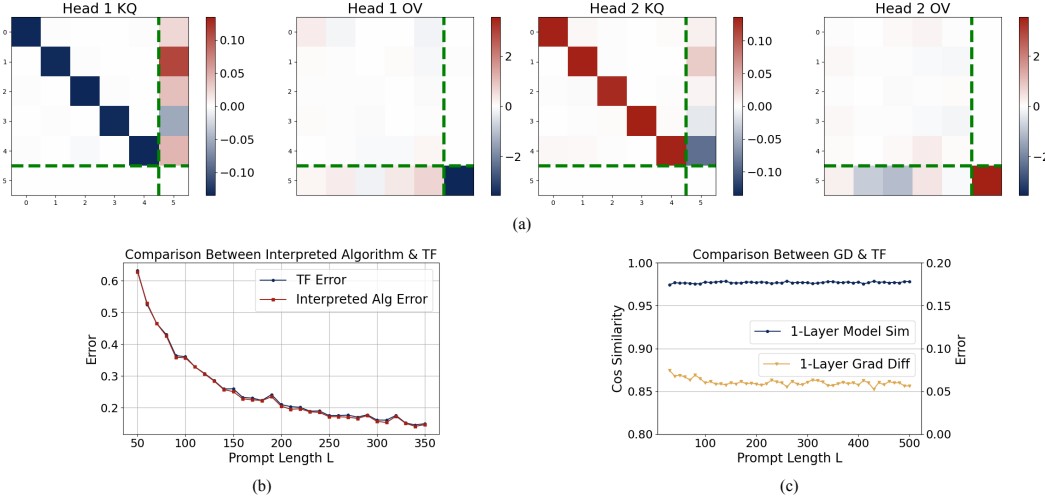

(a)

(b)                                      (c)

Figure 21: Results for a 1-layer, 2-head transformer: (a) visualizations of KQ and OV patterns; (b) performance comparison between the transformer and the interpreted algorithm; (c) alignment between the transformer and the interpreted algorithm.

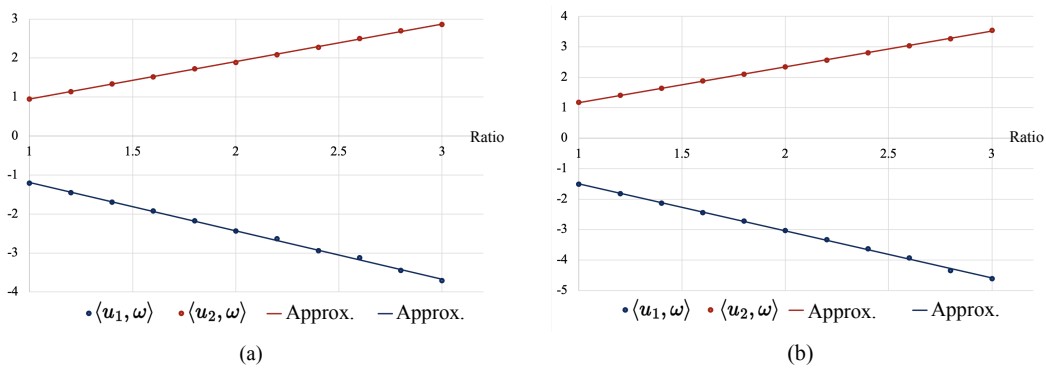

(a)                                      (b)

Figure 22: Results for relationship between covariance and inner products: (a) the approximation of the relationships $\langle u_1, \omega \rangle$ vs. eigenvalue ratios and $\langle u_2, \omega \rangle$ vs. eigenvalue ratios when the initial eigenvalues are constant; (b) the approximation relationship when initial eigenvalues are various.

formance. A finding supports the alignment between interpreted algorithm and transformer by the empirical evidence in Figure 21 (b) and (c).

## C.5   RELATIONSHIP BETWEEN COVARIANCE AND INNER PRODUCTS

In this subsection, we investigate how the eigenvalues $(\lambda_1, \ldots, \lambda_d)$ of the covariance matrix influence the inner products $\langle u_1, \omega \rangle$ and $\langle u_2, \omega \rangle$. We study two cases: a fixed set of eigenvalues and a varying set of eigenvalues. In the first case, the reference eigenvalues are fixed as $\Lambda_0 = (1.0, 0.75, 0.5, 0.75, 1.0)$. We introduce a scaling factor $k$ such that $\Lambda = k \cdot \Lambda_0$. In the second case, the eigenvalues are sampled from $\mathrm{Unif}(\lambda_{\min}, \lambda_{\max})$, where the reference values are $\lambda_{\min} = 0.8$ and $\lambda_{\max} = 1.2$. The scaling factor $k$ is applied to adjust both bounds. For both cases, we vary $k$ from 1 to 3 in increments of 0.2 and conduct experiments using a two-layer transformer.

Across both settings, the KQ and OV circuits remain diagonal, consistent with the findings in Section 4. Examining the learned inner products with respect to the scaling factor $k$, we arrive at the following key result:

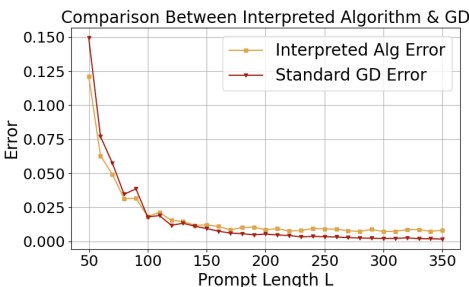

Figure 23: Performance comparison between interpreted algorithm and GD algorithm with the optimal step-size.

> **Observation.** The inner products $\langle u_1, \omega \rangle$ and $\langle u_2, \omega \rangle$ are inversely proportional to the scaling factor $k$.

This observation holds in both the fixed and varying eigenvalue settings. As shown in Figure 22, the relationship between $k$ and the reciprocals of the inner products is approximately linear, thereby validating our observation.

These results highlight a primary connection between the eigenvalues of the covariance matrix and the learned inner products. A deeper investigation into this relationship, including its theoretical underpinnings and implications for generalization, will be an important direction for future work.

## C.6    INTERPRETED ALGORITHM VS. STANDARD GD ALGORITHM

In this subsection, we examine the performance differences between our interpreted algorithm and the standard GD algorithm. Recall that the interpreted algorithm performs iterative updates with step sizes $\eta_x$ and $\eta_y$, which are learned as functions of the number of layers $T$. Consequently, the transformer effectively learns an optimal step size for a given $T$. In other words, the interpreted algorithm adaptively selects optimal step sizes, whereas the standard GD algorithm typically relies on a fixed learning rate that is not tailored to any specific iteration budget. This mismatch often leads to inferior performance for GD under standard settings (see Figure 3 in Section 4). Here, we additionally evaluate a 3-step GD with an optimally tuned learning rate. We search over learning rates in the range $[0.1, 5.0]$ with increments of $0.1$ and identify $1.1$ as the optimal value for the 3-step GD baseline. As shown in Figure 23, GD with this optimal learning rate performs slightly better than the interpreted algorithm, but the gap is small (less than 0.01). Notably, the optimal GD learning rate also depends on the number of iterations, mirroring the dependency observed in our interpreted algorithm.

## C.7    DISCUSSION ON MODEL'S PERFORMANCE ON OUT-OF-DISTRIBUTION TASKS

In this subsection, we evaluate the model's robustness under distribution shift by training on covariance matrices sampled from one eigenvalue distribution and testing on another. Recall that the eigenvalues of the training covariance matrices are drawn from $\text{Unif}(\lambda_{min}, \lambda_{max})$, with $\lambda_{min} = 0.8$ and $\lambda_{max} = 1.2$ in our settings. We train 2-, 3-, and 4-layer transformers using this distribution and then test them on the test sequences sampled from $\mathcal{N}(0, \Sigma_{\text{test}})$. The eigenvalues of covariance matrices $\Sigma_{\text{test}}$ are sampled from alternative uniform distributions. Therefore, the covariance distribution of the training sequences differs from that used during testing. We consider three test-time settings with increasing degrees of shift: large overlap ($\lambda_{min} = 0.9$, $\lambda_{max} = 1.3$), small overlap ($\lambda_{min} = 1.1$, $\lambda_{max} = 1.5$), no overlap ($\lambda_{min} = 1.3$, $\lambda_{max} = 1.7$). The results, shown in Figure 24, indicate that well-trained multi-layer looped transformers exhibit notable tolerance to moderate distribution shifts, though performance degrades as the gap between training and testing distributions widens. In addition, when the shift becomes extreme (e.g., $\lambda_{min} = 3.0$, $\lambda_{max} = 4.0$), the transformer no longer succeeds at the linear regression task.

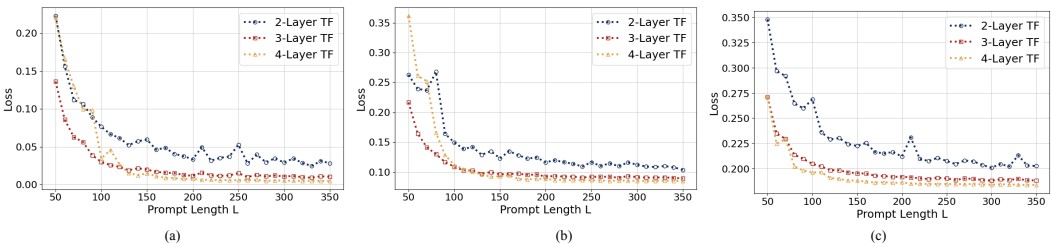

Figure 24: Results for transformer's performance test on the OOD data: (a) performance of large overlap; (b) performance of small overlap; (c) performance of no overlap.

## D THE USE OF LARGE LANGUAGE MODELS (LLMs)

We use LLMs only to polish our writing in the main paper and the appendix.

