# OpenReview forum: "Interpreting Multi-Layer Transformers for In-Context Linear Regression with Varying Covariance"
_ICLR.cc/2026/Conference — Submitted to ICLR 2026_

### Official Review · Reviewer_pBWf · 2025-10-30

**Soundness:** 3
**Presentation:** 2
**Contribution:** 2
**Rating:** 4
**Confidence:** 3

**Summary:**

This paper studies looped multi-layer softmax attention trained to perform in-context linear regression with the in-context covariance varying across sequences. Mechanistic interpretability analysis is conducted to infer the in-context learning algorithm implemented by the trained transformer.

**Strengths:**

- This paper considers in-context linear regression task with varying in-context covariance, which is an interesting and under-explored direction in the literature.
- This paper studies multi-layer attention, making progress beyond the commonly used single-layer setup.

**Weaknesses:**

- The second sentence of the abstract says that

  > "multi-layer transformers substantially outperform single-layer models, demonstrating that depth is critical for robust generalization"

  However, as what I understood and as what the authors elaborate in lines 84 and 213, multi-layer transformers outperforming single-layer ones demonstrates their improved expressivity, rather than improved generalization. Here, the single-layer transformer lacks sufficient expressivity to solve the task, which explains its poorer performance. To demonstrate generalization properties, both single-layer and multi-layer models would need to be expressive enough to fit the training data, with the multi-layer model achieving lower generalization error.

- I am not sure that the setup here should be termed as out-of-distribution (line 48). The training set contains sequences with different in-context covariance matrices, and the test set contains sequences with unseen in-context covariance, but they are still generated according to the same procedure in Section 3.1. This suggests that the test data are in-distribution, not out-of-distribution. The notion of “out-of-distribution” typically refers to a shift in the overall data-generating distribution between training and testing, rather than variation in the in-context token covariance within individual inputs.

- I am confused by the paragraph at lines 301-306: it's unclear whether Algorithm 1 is the algorithm learned by the looped transformer or a standard transformer. From line 196, I understood that Section 4 is about the looped transformer. However, Algorithm 1 has time-varying step sizes, which the authors describe as a feature for standard transformers.

  More broadly, Section 4 frequently refers to Appendix B.1. While cross-referencing supplementary material is not inherently problematic, in this case it interferes with the self-contained nature of the main body and makes the section hard to read smoothly.

**Questions:**

- Could the authors add a discussion on how sensitive the results are to the looped transformer simplification? Tying weights across layers also imposes expressivity constraints. Since expressivity is a key reason why multi-layer transformers outperform single-layer ones in this setup, I wonder how the looped transformer simplification affects the results.
- Why is the in-context learning algorithm reported to be insensitive to the width? Could the authors clarify the rank of the key and query weight matrices in each attention head? If these matrices are low-rank, the expressivity of an attention layer would presumably depend on the width [1,2].
- In line 186, it is unclear which optimizer the authors ended up using, mini batch SGD or Adam. It seems the experiments are conducted with Adam (line 205), then the sentence "Optimization proceeds via mini-batch gradient descent" is misleading.

[1] Amsel, N., Yehudai, G., & Bruna, J. "Quality over Quantity in Attention Layers: When Adding More Heads Hurts." ICLR 2025.

[2] Zhang, Y., Singh, A. K., Latham, P. E., & Saxe, A. M. "Training Dynamics of In-Context Learning in Linear Attention." ICML 2025.

---

> ### Author Response · Authors · 2025-11-20
> **Response to pBWf**
>
> Thanks for your comments. Below, we use A1 for the response to question 1 and so on. We also use A4, A5, and A6 for the responses to weakness 1, 2, and 3, respectively.
>
> **A1**: We have added a performance comparison between the looped and standard transformers and the corresponding discussion in Appendix C.1, using a 2-layer transformer as an example. The results show that their performances are comparable, suggesting that the multi-layer looped transformer is sufficiently expressive to solve the linear regression task under our regime.
> In addition, as shown in the interpretation of the structured original transformer in Appendix C.1, the algorithm learned by the looped transformer is identical to that of the original transformer, except for the differences in the step-size. Therefore, it is reasonable that their performance is closely aligned.
>
> **A2**: Our experiments indicate that the learned top-left d-by-d submatrices of the KQ matrices are full rank. This suggests that the width of attention head should be at least as large as the input dimension $d$. Therefore, without loss of generality, we focus our analysis on the d-by-d case.
> Furthermore, the ICL prediction error remains consistent across different numbers of heads because the inner products $\langle u_1,\omega\rangle$ and $\langle u_2,\omega\rangle$ are identical, which is previously studied by [1]. As a result, the step size in the interpreted algorithm remains unchanged, leading to comparable performance regardless of the number of heads.
>
> **A3**: Thank you for noting this issue. We actually use Adam as the optimizer, which is a version of mini-batch gradient-based algorithm. We have corrected this error in the paper.
>
> **A4**: Thank you for pointing out the ambiguity. Here, we use generalization to refer specifically to the model's ability to solve linear regression tasks under varying input covariance structures. Our goal is to demonstrate that the model generalizes across sequences with different covariance patterns, enabled by its strong expressive capacity. We will revise the abstract to clarify this usage and avoid confusion.
>
> **A5**: Thank you for pointing out this. We have recognized this issue and revised the paper by removing the notion of OOD to avoid confusion. We originally referred to the setting as OOD because the distributions $N(0,\Sigma)$ of the training and testing sequences differ, due to having different covariance matrices.
> But the eigenvalues of these covariance matrices are drawn from the same meta-distribution $Unif(\lambda_{min},\lambda_{max})$.
> In other words, the meta-distributions of the training and testing data are identical. As a result, from a more rigorous perspective, to have OOD, we need to change this meta-distribution of covariance matrices.
>
>
> To address OOD more thoroughly, we now include additional experiments in Appendix Section C.7 that evaluate transformer performance when the test-time covariance matrices are drawn from a different meta-distribution than those used during training, i.e., we change the $\lambda_{min}$ and $\lambda_{max}$ in the test-time.
> In this setup, the difference between the covariance meta-distribution captures the severity of OOD (degree of distributional shift).
> The results show that multi-layer transformers can tolerate moderate shifts in the covariance distribution; however, the ICL prediction error increases as the test-time distribution becomes substantially different from the training distribution.
>
> **A6**: We identified this issue based on the feedback from you and reviewer khMx. In Algorithm 1 of the main paper, the learning process is implemented using a looped transformer. To avoid confusion, we have revised the notation by removing the superscript $t$, as the step size at each iteration remains identical due to weight tying. By the way, the algorithm learned by some structured original transformer also presents the similar content to Algorithm 1 but with the different step-size in each step $t$. We state it in our appendix.
>
> In addition, some empirical results in Section 4.1 are moved to Appendix B.1 due to space limitations; in the final version, we will integrate these results into the main paper because we will have one more page and more space. We have also reduced cross-referencing to Appendix B.1 and added guidance at the end of the section to direct readers to the appendix for further discussion of the learned algorithm, improving overall readability.
>
> ## References
> [1] He, J., Pan, X., Chen, S., and Yang, Z. (2025). In-context linear regression demystified: Training dynamics and mechanistic interpretability of multi-head softmax attention.

---

> > ### Author Response · Authors · 2025-12-01
> > **Summary of our responses**
> >
> > We have summarized the reviewer's questions and our answers below.
> >
> > **Model Architecture:**
> > - *Looped vs. Standard Transformer (Q1):* We addressed concerns about the expressivity of weight-tying. In Appendix C.1, We added a comparison showing that looped and standard transformers achieve comparable performance, confirming that the looped architecture is sufficiently expressive for this task.
> > - *Width Sensitivity (Q2):* We explained that the learned top-left sub-matrices of the KQ matrices are full rank ($d\times d$). The model is insensitive to the number of heads because the specific inner products governing the step sizes ($\langle u,\omega\rangle$) remain identical regardless of the head count, a phenomenon consistent with prior work [1].
> >
> > **Definitions and Descriptions:**
> > - *"Generalization" vs. "Expressivity" (W1):* We have revised the abstract to clarify that "generalization" in our context refers specifically to handling varying covariance structures, which is achieved by the improved model's expressivity by the depth increase.
> > - *Out-of-Distribution (W2):* We agreed that our original setup (same meta-distribution of covariance) is confusing. We removed the OOD label for the standard setup and directly presented our regime in the revised paper. Furthermore, We added new experiments in Appendix C.7 testing true OOD scenarios (shifting the distribution of eigenvalues), showing the model is robust to moderate shifts.
> >
> > **Presentation and Notation:**
> > - *Algorithm 1 Notation (W3):* We fixed the notation in Algorithm 1 to remove layer-specific superscripts (t), aligning it correctly with the looped transformer (constant step sizes).
> > - *Appendix Dependency (W3):* We reduced cross-referencing to Appendix B.1 and added guidance at the end of the section to direct readers to the appendix for further discussion of the learned algorithm, improving overall readability.

---

### Official Review · Reviewer_khMx · 2025-10-30

**Soundness:** 3
**Presentation:** 2
**Contribution:** 3
**Rating:** 6
**Confidence:** 3

**Summary:**

This paper investigates in-context learning (ICL) mechanisms in trained transformers under controlled settings. The observed behaviors align well with existing theoretical constructions.

**Strengths:**

The problem is well-motivated and important. Prior theoretical work largely studies what ICL mechanisms could exist and how to construct them; this paper advances that line by probing trained transformers directly. Although the experiments are conducted in highly controlled settings (which is reasonable), the observations are fairly comprehensive.

**Weaknesses:**

(1) The relationship between the looped transformer and the "original" transformer (without weight tying) is unclear. The preliminary states "We mainly use looped multi-layer transformers with residual links in our study", yet Observations 2, 3, 4 and Section 4.2 appear to concern models without weight tying, as the KQ and OV matrices are indexed by layer t and head h. If the observations are intended to apply to both variants, this should be stated more clearly.

(2) The discussion following Observation 4 seems inconsistent with Claim 2. If each model is trained only for its own depth, then neither the looped nor the standard transformer can implement the purported algorithm.

(3) Given the "divergent" behavior in Figure 4, the statement of Observation 1 may be misleading. The "marginal gains" are not negligible if increasing the number of layers actually increases error. Moreover, Figure 4 seems to show a 4-step TF performing worse than a 3-step TF, which is not the case in Figure 1.

**Questions:**

See above. If these questions are resolved satisfactorily, I'm willing to adjust my score accordingly.

---

> ### Author Response · Authors · 2025-11-20
> **Response to khMx**
>
> Thanks for your comments. Below, we use A1 for the response to question 1 and so on.
>
> **A1**: We apologize for the confusion. All experiments in the main paper and appendix use the looped transformer, except for Appendix C.1, where we compare the looped and original transformer architectures. Because of the shared weights across layers in the looped transformer, the values of ${\omega^t,u_1^t,u_2^t}$ remain identical across layers. We will remove the superscript $t$ in KQ and OV circuits, and highlight this more clearly in the main paper. Note that Claim 1 in Section 4.2 applies to the general case of a transformer’s output, regardless of whether the weights are tied, so we keep the layer superscript to distinguish the different parameters in different layer.
>
> **A2**: A transformer with a fixed number of layers can be interpreted as Algorithm 1 with the same number of iterations. For example, a three-layer looped transformer learns a gradient-based algorithm that operates in three iterations.
> We notice that the argument that "looped transformer learns to implement a gradient-based algorithm" only holds up to $T$ steps, where $T$ is the number of layers of the transformer. More importantly, as in Claim 2, the step-sizes of the gradient-based algorithm, $\eta_x$ and $\eta_y$, while the same in all layers, implicitly depends on the number of total layers $T$. This means that, when we unroll the looped transformer more than one time (e.g., $T$-layer model applied twice), equivalently, we run a gradient-based algorithm (as in Algorithm 1) with more than $T$ steps, (e.g. $2T$ steps). But the learning rates are only optimized for the $T$-step gradient-based algorithm. As a result, when it is applied to $2T$ steps, essentially the learning rates are too large and the algorithm overshoots the global minimum of the least-square problem. This is the reason why we have Observation 4.
>
> To summarize, Claim 2 should be intuitively understood as follows
> - $T$-layer looped transformer learns to implement an $T$-step gradient-based optimization algorithm (Algorithm 1).
> - The learning rates of this gradient-based algorithm are optimized for $T$-step iterations, and thus depends on $T$.
>
> **A3**: The divergent behavior shown in Figure 4 occurs when a transformer trained with a given depth $T$ is evaluated using a different depth $T'$. For example, in Figure 4(a), we use a well-trained three-layer transformer and compute the error between the output at each layer and the ground truth. Consequently, the model performs best when $T=3$. For $T>3$, we extend the model by duplicating the final layer, which is feasible because the looped transformer uses identical KQ and OV circuits at every layer. However, applying the model beyond its trained depth leads to degraded performance, illustrating the divergent behavior. A similar pattern appears in the interpreted algorithm.
>
> In Observation 1, transformers are trained at different depths and then evaluated using their respective optimal depths. As a result, the four-layer transformer outperforms the three-layer model due to its increased expressive power.

---

> > ### Author Response · Authors · 2025-12-01
> > **Summary of our responses**
> >
> > We have summarized the reviewer's questions and our answers below. This review said that he/she is willing to adjust my score accordingly if these questions are resolved satisfactorily.
> >
> > **Model Architecture Clarifications**
> >
> > - *Notion Update (Q1):* We explained that all primary experiments utilize the looped transformer architecture (weight-tied). We have removed these superscripts in the revision to reflect that parameters remain identical across layers in our looped setup in our paper. The superscripts only remain in claim 1 in Section 4.2 (the layer output for a general transformer) and the discussion about standard transformer vs. looped transformer in the Appendix C.1.
> >
> > **Explanations of Contradictions:**
> >
> > - *Algorithm-Depth Dependency (Q2):* We explained the relationship between the transformer depth and the interpreted algorithm, and provided intuitive understanding for Claim 2. An $T$-layer looped transformer learns an $T$-step gradient-based algorithm. Crucially, the learned step sizes ($\eta_x, \eta_y$) are optimized specifically for that finite $T$. If the model is unrolled beyond its training depth (e.g., $2T$), these fixed step sizes become too aggressive, causing the algorithm to overshoot the global minimum (explaining Observation 4).
> >
> > **Experimental Distinctions:**
> >
> > - *Training vs. Unrolling (Q3):* We explained the perceived contradiction between Observation 1 and Figure 4 by distinguishing the setups: ***Observation 1 (Figure 1):*** Compares separate models trained from scratch at different depths (e.g., a 4-layer model outperforms a 3-layer model); ***Divergence (Figure 4):*** Takes a single model trained at depth $T=3$ and forces it to unroll for $T>3$ layers, which can be achieved due to the shared weights in the looped transformer. This degrades performance due to the specific step-size optimization mentioned above.

---

### Official Review · Reviewer_vFTn · 2025-11-01

**Soundness:** 2
**Presentation:** 2
**Contribution:** 2
**Rating:** 4
**Confidence:** 3

**Summary:**

This paper investigates how multi-layer softmax-attention transformers perform in-context linear regression under the challenging setting where the covariate distribution (specifically, the covariance matrix) varies across sequences. The authors demonstrate that depth is critical for robust performance, with multi-layer models substantially outperforming single-layer ones. Through a novel probing methodology, they show that the transformer learns to implement a specific variant of gradient descent, characterized by consistent diagonal structures in the model's weight matrices. The authors further extend their findings to show that with a chain-of-thought-style intermediate step, the transformer can solve in-context instrumental variable (IV) regression.

**Strengths:**

1. The paper moves beyond the common assumption of a fixed covariate distribution, studying a more realistic and challenging scenario with varying covariance across sequences. This directly addresses the critical issue of out-of-distribution generalization in ICL.

2. The paper provides a clear, empirical mechanistic interpretation of the transformer's operation. The identification of consistent diagonal patterns in the Key-Query (KQ) and Output-Value (OV) circuits across layers and heads is a strong contribution, offering a simplified and interpretable view of the learned algorithm.

3. The study is thorough, with extensive experiments and ablations (e.g., varying depth, width, sequence length). The successful application of the interpreted mechanism to the more complex IV regression task strengthens the generalizability and practical relevance of the findings.

**Weaknesses:**

1. The contribution may be perceived as incremental. The paper focuses on softmax transformers but relies on a large-sequence-length regime, which simplifies the analysis and makes the behavior analogous to that of linear attention transformers. A more direct discussion of the necessity and advantages of softmax attention in this specific setting would strengthen the claim of novelty.

2. The parameters in Algorithm 1, such as ηx and ηy, are central to the interpretation. It is unclear whether these are precisely extracted from a trained transformer's weights, and whether the learning rates used in standard gradient descent is optimal for each step/layer in fig3.

3. The algorithm is noted to be non-convergent when applied beyond the trained depth. While it empirically outperforms standard GD within that depth, the paper lacks a theoretical explanation for whythis specific, divergent algorithm is more effective than a convergent one for the in-context learning task

**Questions:**

1. Given the reliance on the large-sequence-length regime for analysis, how do the conclusions hold for very short context lengths where the softmax attention's non-linearity is more critical?

2. How exactly are the step sizes ηx and ηy in Algorithm 1 extracted from the learned model parameters? Are they consistent across layers in the looped transformer, and is there an intuition for why these learned values are effective?

3. Could the authors provide a theoretical intuition for why the non-convergent Algorithm 1 outperforms standard gradient descent within the model's operational depth? Is it effectively performing a form of adaptive preconditioning?

Please refer to the weakness part. If there are any misunderstandings on my part, please point them out, and I will reconsider my evaluation of this work.

---

> ### Author Response · Authors · 2025-11-20
> **Response to the Reviewer vFTn**
>
> Thanks for your comments. Below, we use A1 for the response to question 1 and so on.
>
> **A1**: We think the conclusion will no longer hold and the in-context error will increase, when the sequence length $L$ is too small. This is because the multi-layer transformer characterization in Equation (6) relies on the assumption where $L$ is large. Our analysis treats each transformer layer as an operator acting on the distribution of tokens, and this framework requires the empirical token distribution to closely approximate the population distribution, which becomes valid only when $L$ is sufficiently large. This requirement is consistent with prior work [1, 2, 3], which similarly assumes long input sequences to ensure competitive performance.
>
> **A2**:
> - (1) The step sizes $\eta_x$ and $\eta_y$ are exactly the inner products $\langle u_1, \omega\rangle$ and $\langle u_2, \omega\rangle$ in the KQ and OV circuits, respectively.
> - (2) These values remain consistent across layers due to the shared weight in the looped transformer.
> - (3) The reason why the transformer can learn effective parameters lies in its ability to approximate gradient-based update rules. Specifically, GD with an appropriately chosen learning rate can achieve a small ICL prediction error, and the range of suitable learning rates is relatively broad. At the same time, a multi-layer transformer is expressive enough to represent a family of GD-like algorithms. Consequently, when trained with an appropriate loss function, the transformer can learn an efficient estimator within this GD family, thereby attaining low ICL prediction error. In addition, understanding why the transformer learns particular values for $\eta_x$ and $\eta_y$ requires a dynamic analysis of the training process, which we leave for future work.
>
> **A3**: We believe that Algorithm 1 inherently incorporates the number of layers $T$ in solving the linear regression task. In other words, the optimal step size used by the learned algorithm depends on $T$, introducing an additional constraint. In contrast, standard gradient descent operating within the model’s depth uses a fixed learning rate that does not depend on the number of iterations $T$. We also try to test GD with optimal searched learning rate, and find that the performance is slightly superior to the interpreted algorithm. We add this experiment to our appendix in Section C.6.
>
> ## References
> [1] Ahn, K., Cheng, X., Daneshmand, H., and Sra, S. (2023). Transformers learn to implement preconditioned gradient descent for in-context learning.
>
> [2] He, J., Pan, X., Chen, S., and Yang, Z. (2025). In-context linear regression demystified: Training dynamics and mechanistic interpretability of multi-head softmax attention.
>
> [3] Von Oswald, J., Niklasson, E., Randazzo, E., Sacramento, J., Mordvintsev, A., Zhmoginov, A., and Vladymyrov, M. (2023). Transformers learn in-context by gradient descent

---

> ### Author Response · Authors · 2025-12-01
> **Summary of our responses**
>
> We have summarized the reviewer's questions and our answers below. This reviewer said that he/she will reconsider his/her evaluation of this work after rebuttal.
>
> **Theoretical Assumptions:**
>
> - *Short Context Lengths (Q1):* We acknowledged that our theoretical characterization relies on large sequence lengths $L$ to treat layers as operators on token distributions. The reason is similar to the response for Reviewer 5xqb (Q4). For very small $L$, the approximation of the population distribution fails, likely increasing in-context error. This constraint is consistent with the assumptions in the prior literature.
>
> **Algorithm Interpretation:**
>
> - *Extracting Step Sizes (Q2):* We clarify how step sizes $(\eta_x,\eta_y)$ are extracted as inner products within the KQ and OV circuits and revised our paper to highlight it. These remain consistent across layers due to the looped transformer's shared weights.
> - *Effectiveness (Q2):* We explained the reason of why learned values are effective. We claimed that GD with an appropriately chosen learning rate can achieve a small ICL prediction error. The transformer effectively approximates a GD family of algorithms through the proper loss function training. It locates an efficient estimator (specific step sizes) within this family, thereby attaining low ICL prediction error.
>
> **Comparison to Standard GD:**
>
> - *Performance Intuition (Q3):* We explained that the learned algorithm outperforms standard GD (with fixed rates) because it optimizes step sizes specifically for the model's fixed depth $T$. We also added an experiment in **Appendix C.6** showing that if standard GD is allowed an optimally searched learning rate, it performs slightly better than the interpreted algorithm.

---

### Official Review · Reviewer_5xqb · 2025-11-11

**Soundness:** 3
**Presentation:** 3
**Contribution:** 2
**Rating:** 4
**Confidence:** 3

**Summary:**

Introduction/Motivation and Overview
1. Previous works have shown that transformers are doing gradient descent (GD) during in-context learning (ICL), in synthetic linear regression settings.
2. However, those setups’ learned models are not robust to distribution shifts, because the pre-conditioner matrix is based on the covariance of the data. So they may not work if the data distribution is changed.
3. This work seeks to answer the question: can the transformer solve linear regression with varying covariances? If so, what is the algorithm the transformer learns?
4. ICL ability increases with depth, but stops at a certain depth.
5. This work also finds that in this setting, the transformer learns GD - each layer first performs a linear transformation, then performs a GD step on a linear regression problem using the covariates from the previous layer.
    1. This work shows that the loss of this interpreted algorithm increases beyond a certain number of iterations.
6. Next, this work also studies instrumental variable regression and shows that a transformer trained on instrumental variable regression also performs the same interpreted algorithm.

Setup
1. The x_i are drawn from a Gaussian distribution, where the covariance matrix is fixed within a sequence, and changes across sequences.
2. The test input x_q, within a sequence, may differ from the training distribution.
3. The transformer uses multi-head softmax attention. The causal mask, different from standard attention, makes it so that previous positions don’t attend to the very final position.
4. It is a looped transformer where each attention layer has the same weights.

Interpretability Experiments
1. Observation 1 - performance improves with depth, but depth doesn’t seem useful beyond 3 layers.
2. Observation 2 - attention weights are constant across layers/heads, and form a diagonal matrix.
3. Observation 4 - the transformer, when stacked on top of itself, diverges instead of improving the prediction error.

In section 4.2, the algorithm that the 3-layer looped transformer learns, is determined as follows.
1. Claim 1 - assumes that the sequence length goes to infinity, and gives an interpretation of each layer as moving the activations from one Gaussian distribution to another.
2. Claim 2 - gives a form for the update rule, and also specifies the learning rate.
3. Experimental analysis - they show that the transformer behaves similarly to these update rules.
    1. They compare the following algorithms, (1) the trained 3-layer 2-head softmax transformer, (2) the interpreted algorithm, (3) three-step GD estimator, and (4) ridge regression estimator
    2. Figure 3a shows that the transformer behaves similarly to its interpreted algorithms, i.e. (1) and (2) perform similarly, and are both worse than ridge regression.
    3. Figure 3b shows that, as the length of the sequence increases, the transformer shows more alignment with the interpreted algorithm (measured by the cosine similarity/L2 distance of the input sensitivity vectors, i.e. gradient with respect to input).

They also find that the interpreted algorithm has a divergence in the error as the number of steps increases - this is shown in Figure 4(a) and 4(b).
- Figure 4(b) shows that the interpreted algorithm of the 3-layer transformer performs worse as the number of steps increases.

IV Regression

The instrumental variable regression setting is as follows.
1. Now, the covariate is correlated with the label noise.
2. Thus, the new estimator is as follows. Introduce a new variable z_l that is correlated with x_l, but uncorrelated with the noise.
3. The z_l are sampled randomly from the Gaussian distribution.
4. First, x_l is regressed against z_l to obtain some estimated values of x_l. Then, y_l is regressed against the estimated values of x_l.

They guide the transformer to implement the two-stage estimator for IV regression.
1. The input to the first transformer is (x_i, z_i) for each position.
2. The input to the second transformer is (x_i_hat, y_i) for each position, where x_i_hat is the output of the first transformer.
3. The gradient of the prediction of the first transformer is detached from the input to the second transformer.

Section 5.2 - Interpretation of learned algorithm/results
1. As the sequence length increases, the performance of the transformer converges to that of the 2-stage estimator. Worth noting that at shorter sequences, the 2-stage estimator performs much better. There also still seems to be a slight gap at the end - not sure if they would converge with a further increase in the sequence length, or if there is an asymptotic gap.
2. The second transformer block is essentially solving linear regression where the covariances differ across sequences. Thus, the results from the previous setting (linear regression with varying covariances) allows for predicting the learned algorithm in this setting.

**Strengths:**

- The result on instrumental variable regression shows that the findings of the previous section are relatively robust.

**Weaknesses:**

- One weakness is that a looped transformer is used for the linear regression setting. Thus, it seems perhaps intuitive and not surprising that the error would diverge as the layer is applied multiple times, since the layer is optimized in a way that minimizes the error when it is applied exactly 3 times.
- The architecture seems somewhat hard-coded for the instrumental variable regression setting. There are two transformer blocks, where Block 1 is intended to perform the first stage of the instrumental variable regression estimator, and Block 2 performs the second stage. Additionally, Block 1 does not receive any gradient from Block 2, if I understood correctly.

**Questions:**

1. Could you explain the motivation for your particular choice of casual mask?
2. Regarding observation 1 - in Figure (1a), there actually seems to be a significant improvement from 3 layers to 4 layers, for a large enough prompt length? So this contradicts what is said in the text around lines 212-215.
3. Observation 2 is also confusing.
    1. In lines 178-179, it is mentioned that the attention weights are all the same for each layer.
    2. However, here it is said that the weights are “approximately” the same across layers, in lines 237-239 - this sounds contradictory?
4. In Claim 1, could you give an intuitive explanation of how the assumption that L goes to infinity comes into play?
    1. I assume this is relevant because the transformer is actually bi-directional aside from the last position, and therefore the earlier positions are also affected by the sequence length being infinity.
5. Confusions about the IV regression setting - Section 5.1
    1. Why use only a single-layer transformer for the first block? It sounds possible that more layers would get a better estimate for the x_hat.
    2. In line 416, you state that the “gradient of the sequence is detached”, referring to the gradient of the output of the first transformer block, while in line 422, you state “we train two transformer blocks together”. Are these contradictory?
    3. Is it correct to say that you are heavily enforcing that the transformer should imitate the two-stage estimator for IV regression?
6. In Figure 5c, is there a typo in the title? It currently says “comparison between GD and TF” but perhaps it is intended to be a comparison between the interpreted algorithm and TF.

---

> ### Author Response · Authors · 2025-11-20
> **Response to Reviewer 5xqb**
>
> Thanks for your comments. Below, we use A1 for the response to question 1 and so on.
>
> **A1**: The causal mask is used to reflect the asymmetric structure of the input sequence. Because we manually set $y_q=0$ in the training sequence and require the transformer to predict the value at position $y_q$, it is essential to prevent the query position from attending to itself. This constraint forces the model to infer the query token solely from the remaining $L$ tokens ($L$ is the sequence length). This form of causal masking is standard and has been widely adopted in related work [1, 2, 3]
>
> **A2**: In Figure 1(a), the plotted error is shown on a **log scale**. The absolute difference between the errors of the 3-layer and 4-layer transformers is less than 0.01. Thus, the difference is small and there is no contradiction. We provide the corresponding absolute-scale figure in the Appendix B.1, where the improvement is shown to be negligible.
>
> **A3**: Sorry for the confusion. In line 178-179, we mean that the parameters in the KQ and OV circuits are identical across layers due to the shared weights in looped transformer. In Observation 2, we instead refer to the diagonal elements (excluding the final entry) being approximately equal. For example, if we denote the diagonal elements of the KQ circuit in a given layer as [$u_1, u_2, \dots, u_{d+1}$] are the diagonal elements of KQ circuits in a layer, then the value of $u_1, u_2, \dots, u_{d}$ are approximately the same, which can also be seen in Figure 2.
>
> **A4**: We take $L$ to infinity as a simplifying assumption in our analysis. In particular, when examining how tokens evolve across transformer layers, it is impractical to track individual tokens, as the behavior would depend on the specific sequence length. Instead, we treat a single transformer layer as an operator and study how it transforms the distribution of tokens. This requires the empirical token distribution to approximate the real population distribution, which is ensured by considering the limit $L$ to infinity. In practice, we use a sufficiently large $L$ to approximate this regime. This also explains why the transformer's performance improves as $L$ increases in the early testing stage (when $L$ is small).
>
> **A5**: Thank you for pointing out the ambiguity.
> - (1) We use a single-layer transformer in the first stage because this stage involves a simple linear regression task with a fixed covariance structure. A single-layer transformer is sufficient for this setting, as it can effectively learn a preconditioned gradient descent estimator, which has been extensively studied in prior work [1, 2].
> - (2) During the construction of training sequences for the second stage, we stop the gradient only when generating these sequences; this is done by running the first-stage transformer in inference mode. We have modified the sequence generation part in Section 5 to avoid this confusion.
> - (3) Yes, our goal is to encourage the transformer to imitate a two-stage estimator for IV regression. Since two-stage least squares is the canonical approach for IV estimation, it naturally motivates us to design a two-block architecture. However, it remains unclear whether two transformer blocks trained with the corresponding loss function (Equation 11) can successfully solve the IV task. Moreover, we are interested in understanding what algorithm each stage of the transformer learns. Consequently, our analysis focuses on the two-stage algorithm that the transformer learned and examines whether a multi-layer transformer can handle the more complex ICL settings (e.g., the second stage of the IV task).
>
> **A6**: Thanks for spotting this typo. We have fixed it in our new version.
>
> ## References
> [1] Ahn, K., Cheng, X., Daneshmand, H., and Sra, S. (2023). Transformers learn to implement preconditioned gradient descent for in-context learning.
>
> [2] He, J., Pan, X., Chen, S., and Yang, Z. (2025). In-context linear regression demystified: Training dynamics and mechanistic interpretability of multi-head softmax attention.
>
> [3] Von Oswald, J., Niklasson, E., Randazzo, E., Sacramento, J., Mordvintsev, A., Zhmoginov, A., and Vladymyrov, M. (2023). Transformers learn in-context by gradient descent

---

> > ### Author Response · Authors · 2025-12-01
> > **Summary of our responses**
> >
> > We have summarized the reviewer's questions and our answers below:
> >
> > **Methodological Explanations:**
> >
> > - *Causal mask (Q1):* We clarified that masking the query position is essential to prevent self-attention on the target ($y_q=0$) and force the model to infer the token solely from the context. This aligns with standard practices in related literature [1,2].
> > - *IV Regression Architecture (Q5):* We justified using a single-layer transformer for the first stage as it is sufficient for learning the linear regression estimator. In addition, we also clarified the gradient detachment process (only during sequence generation) and confirmed that the study on IV regression is to show the potential application of our regime in Section 4 and explore whether transformers can imitate two-stage least squares estimators.
> >
> > **Explanations of Contradictions:**
> >
> > - *Layer Performance (Q2):* We explained that the perceived performance gap between 3-layer and 4-layer models in Figure 1(a) is an artifact of the log scale. The absolute difference is negligible (less than 0.01), which we have demonstrated with a new linear-scale plot in the Appendix B1.
> > - *Weights in TF (Q3):* We resolved the confusion between "identical" and "approximate" weights. The parameters are identical across layers (due to the looped architecture), whereas "approximate" referred specifically to the values of diagonal elements in the KQ and OV circuits.
> >
> > **Theoretical Assumptions:**
> >
> > - *Infinite Sequence Length (Q4):* We explained that the assumption $L$ to infinity is a simplifying analytical tool. It allows us to treat a transformer layer as an operator transforming the token distribution (approximating the population) rather than tracking individual tokens, which is widely used in prior literature [1, 2, 3].
> >
> > **Corrections:**
> >
> > - *Figure Typo (Q6):* We corrected the labeling error in the title of Figure 5c.

---

### Author Response · Authors · 2025-11-20

We sincerely thank the reviewers for their insightful comments and constructive feedback. We have updated the paper accordingly and highlighted all modifications in red font for easy identification.

---

### Author Response · Authors · 2025-12-01

We sincerely thank the Area Chair for reviewing our paper again. We have provided a summary of the reviewers' questions and our responses in the individual comment threads for your reference.

---

### Meta-Review · Area_Chair_fzHY · 2025-12-31

**Summary:**

This paper studies how multi-layer softmax-attention transformers perform in-context linear regression, where the covariance matrix of input changes across sequence. The main observation is that multi-layer transformers substantially outperform single-layer models, and the proposed interpretation is that transformers implement a variant of gradient descent with hyperparameters dependent on model depth and data distribution. The reviewers had concerns on the choice of looped architecture and some reviewers were confused about the divergent behavior and the "time-dependent" aspect of the algorithm. The author responses addressed some of these concerns. However, the concerns that looped transformer seems to be hard-coding the proposed algorithm in some sense and the novelty of the work were not fully addressed.

**Reviewer Concerns:**

There were some major confusions among the reviewers on the divergent behavior and the "time-dependent" aspect of the algorithm. The authors provided more explanations and in particular fixed notations for the latter problem. Reviewers should change their mind on the "time-dependent" issue although they may still worry about the divergent behavior.

Other main concerns include that looped transformer seems to be hard-coding the proposed algorithm and that the paper is a bit incremental. These are not well addressed.

**Reviewer Scores:**

5xqb: questions are answered, weaknesses mostly remain, may stay at 4 or change to 5
vFTn: most concerns answered although some weaknesses remain, likely change to 5
khMx: questions are answered but all results are on looped transformer, I'm not sure whether that is important to the reviewer. already most positive at 6
pBWf: most concerns answered although some weaknesses remain, may stay at 4 or change to 5.

---

### Decision · Program_Chairs · 2026-01-26

Reject